# CXXC-finger protein 1 associates with FOXP3 to stabilize homeostasis and suppressive functions of regulatory T cells

Xiaoyu Meng[1,2,3†], Yezhang Zhu[4,5†], Kuai Liu[1,2,3†], Yuxi Wang[6], Xiaoqian Liu[1,2,3], Chenxin Liu[2], Yan Zeng[1,2,3], Shuai Wang[1,2,3], Xianzhi Gao[1,2,3], Xin Shen[7], Jing Chen[8], Sijue Tao[6], Qianying Xu[2], Linjia Dong[9], Li Shen[10,11]*, Lie Wang[1,2,3,6,12]*

[1]Institute of Immunology and Bone Marrow Transplantation Center, The First Affiliated Hospital, Zhejiang University School of Medicine, Hangzhou, China; [2]Zhejiang University School of Medicine, Hangzhou, China; [3]Liangzhu Laboratory, Zhejiang University Medical Center, Hangzhou, China; [4]Department of Hematology, Tongji Hospital, School of Medicine, Tongji University, Shanghai, China; [5]Shanghai Key Laboratory of Signaling and Disease Research, Frontier Science Center of Stem Cell Research, National Stem Cell Translational Resource Center, School of Life Sciences and Technology, Tongji University, Shanghai, China; [6]Laboratory Animal Center, Zhejiang University, Hangzhou, China; [7]Co-Facility Center, Zhejiang University School of Medicine, Hangzhou, China; [8]Department of Gastrointestinal Surgery, The Second Affiliated Hospital, Zhejiang University School of Medicine, Hangzhou, China; [9]School of Basic Medical Sciences and Forensic Medicine, Hangzhou Medical College, Hangzhou, China; [10]MOE Key Laboratory of Biosystems Homeostasis & Protection and Zhejiang Provincial Key Laboratory for Cancer Molecular Cell Biology, Life Sciences Institute, Zhejiang University, Hangzhou, China; [11]Department of Orthopedics Surgery, The Second Affiliated Hospital, School of Medicine, Zhejiang University, Hangzhou, China; [12]Future Health Laboratory, Innovation Center of Yangtze River Delta, Zhejiang University, Jiaxing, China

*For correspondence:
li_shen@zju.edu.cn (LS);
wanglie@zju.edu.cn (LW)

†These authors contributed equally to this work

## eLife Assessment

This study presents **important** findings on the role of CXXC-finger protein 1 in regulatory T cell gene regulation and function. The evidence supporting the authors' claims is **convincing**, with mostly state-of-the-art technology. The work will be of relevance to immunologists interested in regulatory T cell biology and autoimmunity.

**Abstract** FOXP3-expressing regulatory T ($T_{reg}$) cells play a pivotal role in maintaining immune homeostasis and tolerance, with their activation being crucial for preventing various inflammatory responses. However, the mechanisms governing the epigenetic program in $T_{reg}$ cells during their dynamic activation remain unclear. In this study, we demonstrate that CXXC-finger protein 1 (CXXC1) interacts with the transcription factor FOXP3 and facilitates the regulation of target genes by modulating H3K4me3 deposition. *Cxxc1* deletion in $T_{reg}$ cells leads to severe inflammatory disease and spontaneous T cell activation, with impaired immunosuppressive function. As a transcriptional regulator, CXXC1 promotes the expression of key $T_{reg}$ functional markers under steady-state conditions,

which are essential for the maintenance of T$_{reg}$ cell homeostasis and their suppressive functions. Epigenetically, CXXC1 binds to the genomic regulatory regions of T$_{reg}$ program genes in mouse T$_{reg}$ cells, overlapping with FOXP3-binding sites. Given its critical role in T$_{reg}$ cell homeostasis, CXXC1 presents itself as a promising therapeutic target for autoimmune diseases.

## Introduction

Regulatory T (T$_{reg}$) cells are a distinct subset of CD4$^+$ T cells that play a critical role in maintaining immune homeostasis and self-tolerance by suppressing excessive or aberrant immune responses to foreign or self-antigens (*Josefowicz et al., 2012*; *Sakaguchi et al., 2008*; *Singer et al., 2014*). These cells can be further categorized into thymus-derived regulatory T cells (tT$_{reg}$ cells), periphery-derived T$_{reg}$ cells (pT$_{reg}$ cells), and induced T$_{reg}$ cells (iT$_{reg}$ cells) (*Ohkura and Sakaguchi, 2020*). They uniquely express the transcription factor FOXP3, a member of the forkhead winged-helix family, which is essential for T$_{reg}$ cell lineage commitment and suppressive function (*Littman and Rudensky, 2010*; *Sakaguchi et al., 2013*). Deletion or mutation of the *Foxp3* gene leads to a range of immunological disorders, including allergies, immunopathology, and autoimmune diseases in both mice and humans (*Fontenot et al., 2003*; *Van Gool et al., 2019*).

It is well established that FOXP3 recruits various cofactors to form complexes that either promote or repress the expression of downstream genes, with histone and DNA modifications playing pivotal roles in this process. FOXP3 can activate or repress the transcription of key regulators of T$_{reg}$ cell activation and function by recruiting the histone acetyltransferases or histone deacetylases (*Katoh et al., 2011*; *Wang et al., 2015*). Notably, FOXP3-bound sites exhibit enrichment of H3K27me3, a modification essential for FOXP3-mediated repressive chromatin remodeling under inflammatory conditions (*Arvey et al., 2014*). However, the direct role of FOXP3 as a transcriptional activator through interactions with epigenetic regulators, particularly via modulation of H3K4 trimethylation, remains poorly documented.

In mammals, six proteins have been identified that catalyze H3K4 methylation. These proteins contain the SET domain and include MLL1 (KMT2A), MLL2 (KMT2B), MLL3 (KMT2C), MLL4 (KMT2D), SETD1A, and SETD1B (*Ruthenburg et al., 2007*; *Takahashi et al., 2011*). For example, MLL1 plays a critical role as an epigenetic regulator in T$_{reg}$ cell activation and functional specialization (*Wang et al., 2024*). Additionally, Placek et al. demonstrated that MLL4 is essential for T$_{reg}$ cell development by catalyzing H3K4me1 at distant unbound enhancers through chromatin looping (*Placek et al., 2017*). H3K4me3, which is enriched at the transcription start site (TSS) and the CpG island (CGI), converts chromatin into active euchromatin by recruiting activating factors (*Yang et al., 2021*). CXXC-finger protein 1 (CXXC1, also known as CFP1), which contains a SET1 interaction domain (SID), is required for binding to the histone H3K4 methyltransferases SETD1A and SETD1B (*Tate et al., 2010*). Previous studies have demonstrated that CXXC1 plays a crucial role in regulating promoter patterns during T cell maturation, mediating GM-CSF-derived macrophage phagocytosis, directing TH17 cell differentiation, and modulating the function of ILC3 cells during aging by regulating H3K4me3 modifications (*Cao et al., 2016*; *Hui et al., 2018*; *Lin et al., 2019*; *Shen et al., 2023*). Despite the well-documented role of CXXC1 in various immune effector cells, its role in T$_{reg}$ cells remains unclear.

Here, we demonstrate that CXXC1 interacts with FOXP3 and enhances the expression of FOXP3 target genes. T$_{reg}$ cell-specific deletion of *Cxxc1* triggers systemic autoimmunity, accompanied by multiorgan inflammation, ultimately resulting in early-onset fatal inflammatory disease in mice. *Cxxc1*-deficient T$_{reg}$ cells exhibit a disadvantage in proliferation and homeostasis, even in non-inflammatory mice where coexisting wild-type (WT) T$_{reg}$ cells were present. Moreover, *Cxxc1*-deficient T$_{reg}$ cells display intrinsic defects in the expression of key suppression molecules, including CTLA-4, CD25, ICOS, and GITR. Mechanistic investigations further indicate that CXXC1 serves as an essential cofactor for FOXP3 by maintaining H3K4me3 modifications at critical genes involved in T$_{reg}$ cell function.

# Results

## FOXP3 binds regulatory loci primed for activation and repression in T<sub>reg</sub> cells

FOXP3-mediated gene expression is well recognized, with several studies highlighting its dual role as both a transcriptional activator and repressor (*Fu et al., 2012*; *Marson et al., 2007*; *Ohkura et al., 2012*; *Ono et al., 2007*; *Zheng et al., 2007*). However, the mechanisms by which FOXP3 regulates T<sub>reg</sub>-specific gene transcription via epigenetic modifications remain incompletely understood. To investigate these mechanisms, we employed CUT&Tag to generate genome-wide H3K4me3 maps in T<sub>reg</sub> cells. To complement this, we compared our data with an H3K27me3 ChIP-seq dataset from *Wei et al., 2009* focusing on previously identified FOXP3-bound loci (*Konopacki et al., 2019*). This integrative analysis allowed us to identify FOXP3-dependent genes associated with either H3K4me3 (indicative of transcriptional activation) or H3K27me3 (indicative of repression) deposition. These findings provide insight into how FOXP3 modulates T<sub>reg</sub> cell function through epigenetic modifications.

As expected, H3K4me3 was enriched at gene promoters (*Figure 1—figure supplement 1A, B*). A Venn diagram revealed overlap between FOXP3-binding sites and H3K4me3 peaks, with minimal overlap with H3K27me3 peaks (*Figure 1A, B*). The overlapping regions between FOXP3-binding sites and H3K4me3 or H3K27me3 peaks were predominantly located at promoters (*Figure 1A, B*). To elucidate FOXP3's potential role as an epigenetic regulator, we compared H3K4me3 levels between FOXP3-positive T<sub>reg</sub> cells and FOXP3-negative conventional T cells (T<sub>conv</sub>). Consistent with our hypothesis, H3K4me3 abundance was higher at T<sub>reg</sub>-specific gene loci (e.g., *Tnfrsf18*, *Ctla4*, *Il2ra*, and *Nt5e*) in T<sub>reg</sub> cells compared to T<sub>conv</sub> cells (*Figure 1C*, *Figure 1—figure supplement 1C*). Notably, the selection of these loci was guided by prior studies identifying genes specifically associated with T<sub>reg</sub> cell function (*Hill et al., 2007*). This pattern of epigenetic remodeling robustly supports a model in which FOXP3 orchestrates T<sub>reg</sub>-specific transcriptional programs by selectively recruiting H3K4 trimethylation machinery to key regulatory gene promoters.

To further characterize these modifications, we clustered promoters into four groups based on the enrichment of H3K4me3 and H3K27me3. Clusters 1 and 3 showed strong enrichment of H3K4me3; Cluster 2 was enriched with H3K27me3; and Cluster 4 showed weak enrichment of both modifications. FOXP3 preferentially bound to the promoters of Clusters 1 and 3, which displayed high H3K4me3 levels (*Figure 1D*). Correspondingly, genes in these clusters exhibited high transcription levels, as shown by the reanalysis of previously published RNA-sequencing (RNA-seq) data (*Oh et al., 2017*; *Figure 1E*). In contrast, genes with H3K27me3 enrichment at their promoters were transcribed at low levels.

Gene Ontology (GO) analysis of these four clusters revealed distinct functional roles. Cluster 1 was enriched in genes involved in mRNA processing, covalent chromatin modification, and histone modification, while Cluster 3 was enriched in genes related to DNA repair and mitochondrion organization (*Figure 1F*). Cluster 2, enriched with H3K27me3, was associated with the pattern specification process, whereas Cluster 4 showed no correlation with T<sub>reg</sub> cells. Notably, signature T<sub>reg</sub> cell genes such as *Tnfrsf18*, *Nrp1*, *Stat5a*, *Lag3*, *Icos*, and *Pdcd1* were enriched in Clusters 1 and 3, showing strong H3K4me3 marks (*Figure 1—figure supplement 1D*). Conversely, genes like *Hic1*, *Trp73*, and *Rnf157*, associated with inflammatory responses, were enriched for H3K27me3 in Cluster 2 (*Figure 1—figure supplement 1D*). These findings collectively support the conclusion that FOXP3 contributes to transcriptional activation in T<sub>reg</sub> cells by promoting H3K4me3 deposition at target loci, while also regulating gene expression directly or indirectly through other epigenetic modifications.

## CXXC1 interacts with FOXP3 and binds H3K4me3-enriched sites in T<sub>reg</sub> cells

We conducted an enrichment analysis of known motifs at the overlapping peaks of FOXP3 ChIP-seq and H3K4me3 CUT&Tag in T<sub>reg</sub> cells to identify epigenetic factors that directly interact with FOXP3 to mediate chromatin remodeling and transcriptional reprogramming. Motif analysis of the overlapping peaks between FOXP3-binding sites and regions enriched in H3K4me3 revealed that, in addition to transcription factors, the most abundant motif associated with H3K4me3 was the epigenetic factor CXXC1 (*Figure 2—figure supplement 1A*). To investigate this further, we performed CUT&Tag for endogenous CXXC1 in T<sub>reg</sub> cells to examine the genome-wide co-occupancy of CXXC1 and FOXP3.

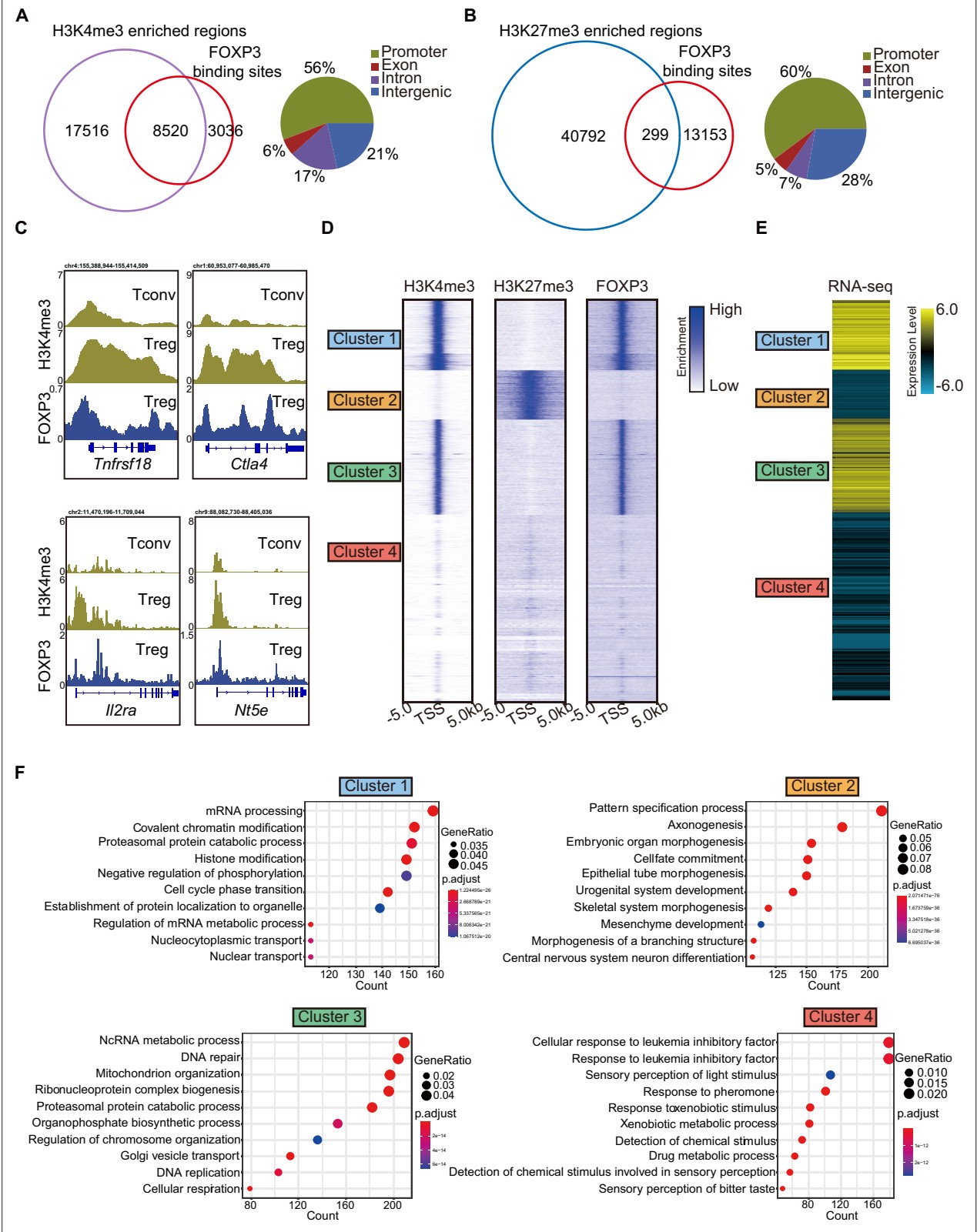

**Figure 1.** H3K4me3 is required for FOXP3-dependent gene activation in T_reg cells. (**A**) Venn diagram showing overlap of H3K4me3-enriched regions (this study) and FOXP3-binding sites *Konopacki et al., 2019* in sorted CD4+YFP+T_reg cells (left). Genomic distribution of overlapped peaks (right). Note that the overlapped peaks are predominantly enriched at promoters. (**B**) Venn diagram showing overlap of H3K27me3-enriched regions (*Wei et al., 2009*) and FOXP3-binding sites in T_reg cells (left). Genomic distribution of overlapped peaks (right). Note that the overlapped peaks are

*Figure 1 continued on next page*

*Figure 1 continued*

predominantly enriched at promoters. (**C**) Representative genome browser view showing the enrichments of H3K4me3 and FOXP3 in $T_{conv}$ or $T_{reg}$ cells. (**D**) Heatmap showing enrichment of H3K27me3, H3K4me3, and FOXP3 surrounding the transcription start site (TSS). Unsupervised k-means clustering was conducted on H3K27me3 and H3K4me3 signals. (**E**) Heatmap showing gene expression levels in $T_{reg}$ cells (RNA-sequencing [RNA-seq] data was obtained from *Oh et al., 2017*). The clusters were consistent as in **C**. (**F**) Gene Ontology (GO) pathway analysis of different clusters.

The online version of this article includes the following figure supplement(s) for figure 1:

**Figure supplement 1.** H3K4me3 is required for FOXP3-dependent gene activation in $T_{reg}$ cells.

Over half of these CXXC1-binding sites were located at promoter regions (*Figure 2—figure supplement 1B*). Additionally, CXXC1 exhibited strong binding at TSS and CGIs (*Figure 2A*, *Figure 2—figure supplement 1C*). As illustrated by the Venn diagram (*Figure 2B*), more than half of the FOXP3-bound genes and H3K4me3-enriched genes were also bound by CXXC1. Similarly, more than half of CXXC1 peaks were overlapped with FOXP3 peaks (*Figure 2—figure supplement 1D*). Furthermore, the CXXC1- and FOXP3-specific binding sites also demonstrated modest binding of FOXP3 and CXXC1, respectively (*Figure 2C*). These findings indicate that FOXP3 and CXXC1 share a substantial number of target genes in $T_{reg}$ cells. To confirm this interaction, we further validated the reciprocal immuno-precipitation of both endogenous and exogenous CXXC1 and FOXP3 (*Figure 2D*, *Figure 2—figure supplement 1E*). An immunofluorescence assay revealed predominant colocalization of CXXC1 with FOXP3 in the nucleus (*Figure 2E*). Overall, these results suggest that CXXC1 primarily functions as a coactivator of FOXP3-driven transcription in $T_{reg}$ cells.

## Complete ablation of *Cxxc1* in $T_{reg}$ cells leads to a fatal autoimmune disease

To investigate the role of CXXC1 in $T_{reg}$ cell homeostasis and function, we generated $Foxp3^{YFP-Cre}Cxxc1^{fl/fl}$ mice (conditional knockout [cKO] mice) by crossing $Cxxc1^{fl/fl}$ with $Foxp3^{YFP-Cre}$ (*Rubtsov et al., 2008*) mice, thereby specifically deleting *Cxxc1* in $T_{reg}$ cells. The effective depletion of *Cxxc1* in $T_{reg}$ cells was confirmed through quantitative PCR (qPCR) and western blotting (*Figure 3—figure supplement 1A*). Notably, cKO mice appeared normal at birth but later exhibited spontaneous mortality starting around 3 weeks of age (*Figure 3A*). Deletion of *Cxxc1* in $T_{reg}$ cells led to the development of severe inflammatory disease, characterized by reduced body size, stooped posture, crusting of the eyelids, ears, and tail, and skin ulceration, particularly on the head and upper back (*Figure 3B, C*). Additionally, cKO mice developed extensive splenomegaly and lymphadenopathy (*Figure 3D*). $Foxp3^{YFP-Cre}Cxxc1^{fl/fl}$ mice exhibited elevated serum levels of anti-dsDNA autoantibodies and IgG, along with a modest increase in IgE concentration (*Figure 3E*, *Figure 3—figure supplement 1B*). Histopathological analysis revealed massive lymphocyte and myeloid cell infiltration in the skin, lungs, liver sinusoids, and colon mucosa (*Figure 3F*). In full agreement with the aforementioned severe autoimmune diseases, $Foxp3^{YFP-Cre}Cxxc1^{fl/fl}$ mice had decreased percentages and numbers of $CD4^+$ $Foxp3^+$ $T_{reg}$ cells in small intestine lamina propria (LPL), liver, and lung (*Figure 3G*, *Figure 3—figure supplement 1C*). Moreover, cKO mice displayed an increase in $CD8^+$ T cell percentages (*Figure 3—figure supplement 1D*), along with a marked rise in cells exhibiting an effector/memory phenotype (CD44hi CD62Llo) (*Figure 3H*). Furthermore, T cells from cKO mice produced elevated levels of IFN-γ, IL-17, and IL-4 in $CD4^+$ $YFP^-$ T cells, as well as increased IFN-γ production in $CD8^+$ T cells (*Figure 3I*, *Figure 3—figure supplement 1E, F*). These phenotypes closely resembled those observed in *Foxp3*-deficient mice (*Fontenot et al., 2003*) or mice with depleted $T_{reg}$ cells (*Kim et al., 2007*), suggesting a deficiency in immune suppression.

## CXXC1 is necessary for the maintenance of $T_{reg}$ cell suppressive activity

Despite the development of severe autoimmune disease, we observed an increase in both the absolute number and percentage of $FOXP3^+$ $T_{reg}$ cells in the lymph nodes (*Figure 4—figure supplement 1A*). The expression level of the FOXP3 protein was only slightly altered in *Cxxc1*-deficient $T_{reg}$ cells (*Figure 4—figure supplement 1B*). In an in vitro suppression assay, $T_{reg}$ cells from $Foxp3^{YFP-Cre}Cxxc1^{fl/fl}$ and WT mice exhibited similar suppressive effects on naive T (Tn) cell proliferation (*Figure 4—figure supplement 1C*). The expression of the hallmark $T_{reg}$ cell marker CTLA-4 showed a modest increase in *Cxxc1*-deficient $T_{reg}$ cells compared to WT $T_{reg}$ cells, while the expression of GITR remained unchanged

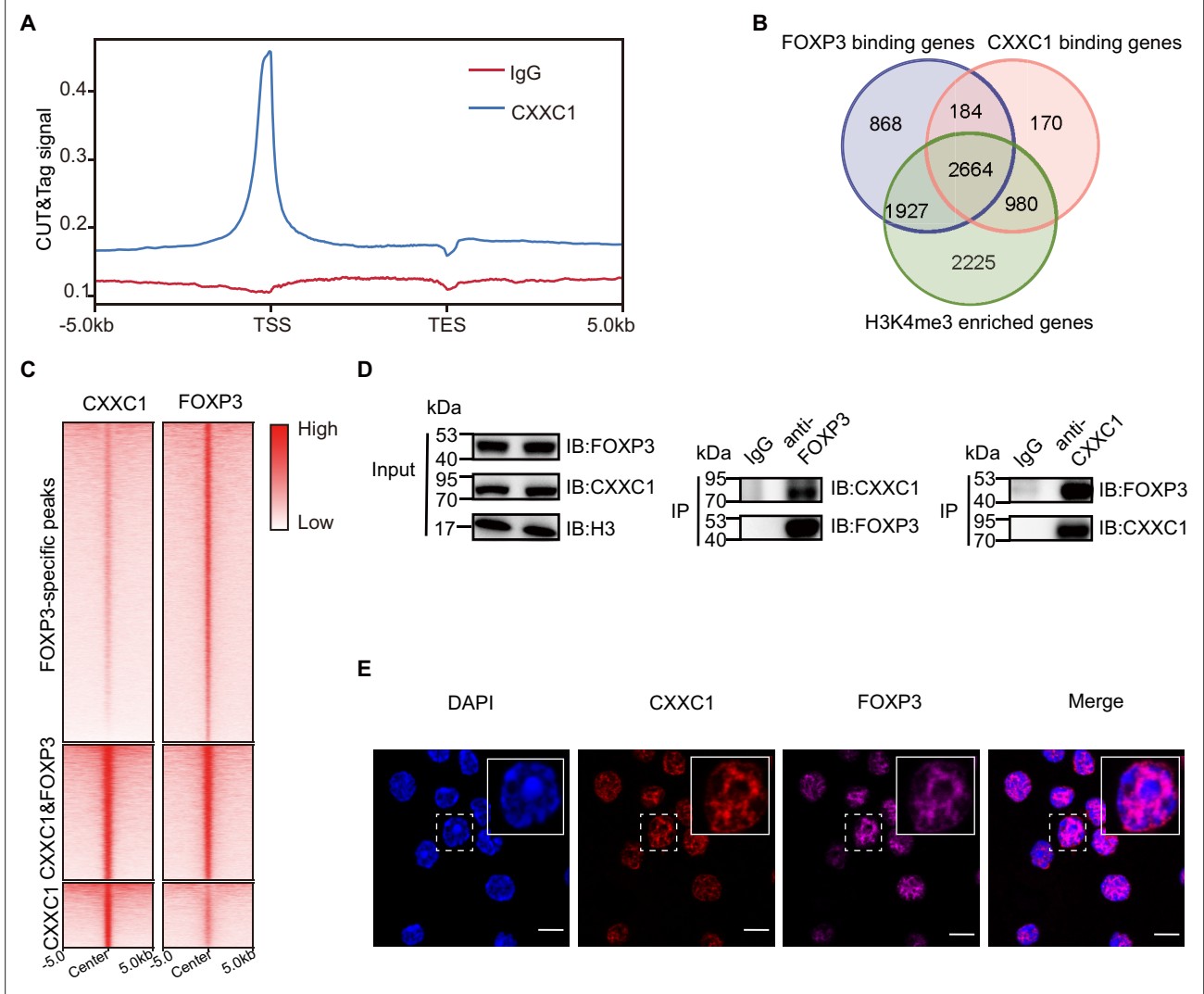

**Figure 2.** CXXC1 interacts with FOXP3 in T_reg cell. (**A**) Average CXXC1 CUT&Tag signals around genes in T_reg cells. IgG was used as the control. (**B**) Venn diagrams showing the overlap of FOXP3-binding genes, CXXC1-binding genes, and H3K4me3-enriched genes in T_reg cells. Genes covered by FOXP3-binding sites, CXXC1-binding sites, or exhibited high H3K4me3 levels at promoters were defined as FOXP3-bound genes, CXXC1-bound genes, or H3K4me3-enriched genes. (**C**) Heatmaps showing FOXP3 ChIP-seq and CXXC1 CUT&Tag signals at indicated regions. (**D**) Interaction between FOXP3 and CXXC1 was assessed by co-IP (forward and reverse) using T_reg cell lysates. (**E**) Immunofluorescence for FOXP3 and CXXC1 colocalization in T_reg cells. Scale bars, 2 μm.

The online version of this article includes the following source data and figure supplement(s) for figure 2:

**Source data 1.** File containing labeled original western blots for *Figure 2*.

**Source data 2.** Original gel image files for western blot analysis displayed in *Figure 2*.

**Figure supplement 1.** FOXP3 interacts with CXXC1.

**Figure supplement 1—source data 1.** File containing labeled original western blots for *Figure 2—figure supplement 1*.

**Figure supplement 1—source data 2.** Original gel image files for western blot analysis displayed in *Figure 2—figure supplement 1*.

(*Figure 4—figure supplement 1D*). To further assess the suppressive capacity of *Cxxc1*-deficient T_reg cells in vivo, we employed the experimental autoimmune encephalomyelitis (EAE) model. Naive CD4+ T cells from 2D2 mice were co-transferred with T_reg cells from either *Foxp3*^YFP-Cre or *Foxp3*^YFP-Cre*Cxxc1*^fl/fl mice into *Rag1*^−/− recipients, and EAE was induced in these recipient mice. Mice that received only naive CD4+ T cells from 2D2 mice developed more severe EAE symptoms (*Figure 4A*). The addition of WT T_reg cells from *Foxp3*^YFP-Cre mice slightly mitigated EAE progression and reduced Th17 cells in the spinal cord (*Figure 4A–D*). In contrast, *Foxp3*^YFP-Cre*Cxxc1*^fl/fl T_reg cells failed to suppress EAE

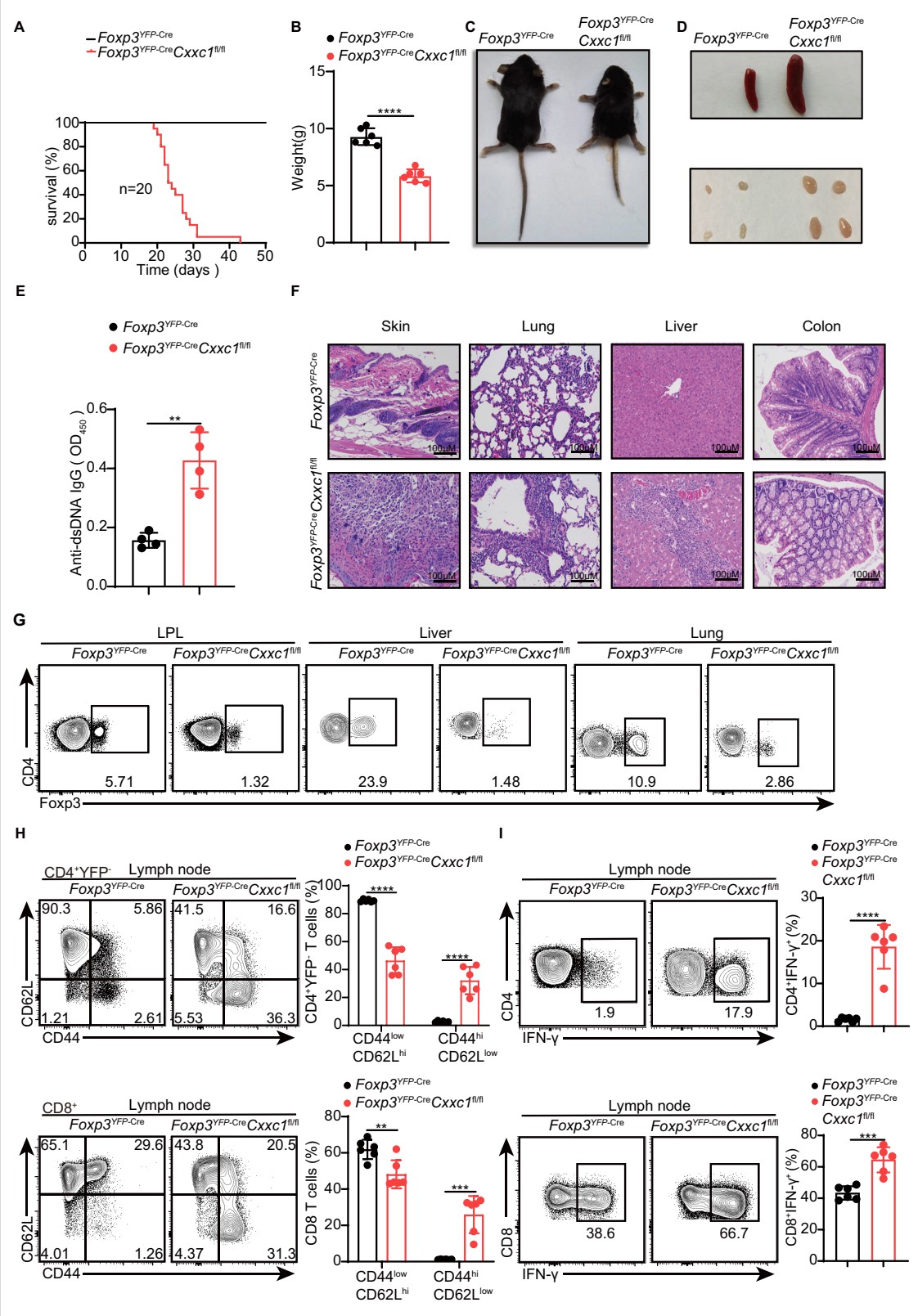

**Figure 3.** *Foxp3*[YFP-Cre]*Cxxc1*[fl/fl] mice spontaneously develop a fatal early-onset inflammatory disorder. (**A**) Survival curves of *Foxp3*[YFP-Cre] (black line) and *Foxp3*[YFP-Cre] *Cxxc1*[fl/fl] (red line) mice (*n* = 20). (**B**) Gross body weight of *Foxp3*[YFP-Cre] and *Foxp3*[YFP-Cre]*Cxxc1*[fl/fl] mice (*n* = 6). (**C**) A representative image of *Foxp3*[YFP-Cre] and *Foxp3*[YFP-Cre]*Cxxc1*[fl/fl] mice. (**D**) Representative images showing the spleen and peripheral lymph nodes from *Foxp3*[YFP-Cre] and *Foxp3*[YFP-Cre]*Cxxc1*[fl/fl] mice. (**E**) ELISA quantification of anti-dsDNA IgG in the serum of *Foxp3*[YFP-Cre] and *Foxp3*[YFP-Cre]*Cxxc1*[fl/fl] mice (*n* = 4). (**F**) Hematoxylin and

*Figure 3 continued on next page*

*Figure 3 continued*

eosin staining of the skin, lung, liver, and colon from *Foxp3*[YFP-Cre] and *Foxp3*[YFP-Cre]*Cxxc1*[fl/fl] mice (scale bar, 100 μm). (**G**) Representative flow cytometry plots of CD4[+] Foxp3[+] T$_{reg}$ cells isolated from the small intestinal lamina propria (LPL), liver, and lung of *Foxp3*[YFP-Cre] and *Foxp3*[YFP-Cre] *Cxxc1*[fl/fl] mice. (**H**) Flow cytometry analysis of CD62L and CD44 expression on peripheral lymph node CD4[+]YFP[−] and CD8[+]T cells from *Foxp3*[YFP-Cre] and *Foxp3*[YFP-Cre]*Cxxc1*[fl/fl] mice (left). Right, frequency of CD44[low]CD62L[hi] and CD44[hi]CD62L[low] population in CD4[+]YFP[−] or CD8[+] T cells (*n* = 6). (**I**) Lymph node cells from *Foxp3*[YFP-Cre] and *Foxp3*[YFP-Cre]*Cxxc1*[fl/fl] mice were stimulated ex vivo with PMA + ionomycin for 4 hr and analyzed for IFN-γ expressing in CD4[+] YFP[−] or CD8[+] T cells using flow cytometry (left). Right, percentages of IFN-γ[+]CD4[+] YFP[−] or IFN-γ[+]CD8[+] T cells in the lymph nodes of *Foxp3*[YFP-Cre] and *Foxp3*[YFP-Cre]*Cxxc1*[fl/fl] mice (*n* = 6). All mice analyzed were 18–20 days old unless otherwise specified. Error bars show mean ± SD. The log-rank survival curve was used for survival analysis in A, and unpaired *t*-test or multiple unpaired *t*-test were used for statistical analyses in **B, E, G– I** (**p < 0.01, ***p < 0.001, ****p < 0.0001). The flow cytometry results are representative of three independent experiments.

The online version of this article includes the following source data and figure supplement(s) for figure 3:

**Source data 1.** Original source data for graphs displayed in *Figure 3*.

**Figure supplement 1.** Disrupted immune homeostasis in *Foxp3*[YFP-Cre]*Cxxc1*[fl/fl] mice.

**Figure supplement 1—source data 1.** File containing labeled original western blots for *Figure 3—figure supplement 1*.

**Figure supplement 1—source data 2.** Original gel image files for western blot analysis displayed in *Figure 3—figure supplement 1*.

**Figure supplement 1—source data 3.** Original source data for graphs displayed in *Figure 3—figure supplement 1*.

(*Figure 4A–D*), and the cKO mice showed a reduction in T$_{reg}$ cell frequency in central nervous system (CNS) tissues (*Figure 4E*). Finally, we examined the role of CXXC1 in T$_{reg}$ cell-mediated suppression using T cell transfer-induced colitis, in which naive T cells were transferred to *Rag1*[−/−] recipients either alone or together with WT or *Foxp3*[YFP-Cre]*Cxxc1*[fl/fl] T$_{reg}$ cells. The transfer of naive T cells led to weight loss and intestinal pathology in recipient mice (*Figure 4F, G*). Mice receiving WT T$_{reg}$ cells continued to gain weight (*Figure 4F*), whereas those that received T$_{reg}$ cells from cKO mice were unable to prevent colitis and exhibited a reduced percentage of T$_{reg}$ cells (*Figure 4F–H*). These findings underscore the critical role of CXXC1 in maintaining T$_{reg}$ cell function in vivo.

## T$_{reg}$ cell lineage homeostasis and proliferation depend upon CXXC1

T$_{reg}$ cells harbor a diverse T cell receptor (TCR) repertoire, which likely plays a critical role in their immune suppression function (*Dikiy and Rudensky, 2023*; *Shevyrev and Tereshchenko, 2019*; *Zagorulya et al., 2023*). To explore the role of CXXC1 in T$_{reg}$-mediated suppression, we performed single-cell RNA sequencing (scRNA-seq) combined with TCR sequencing (TCR-seq) on CD4[+]YFP[+] T$_{reg}$ cells isolated from mouse lymph nodes. After quality control and removal of doublets, 18,577 cells were retained for further analysis. Through unsupervised clustering and uniform manifold approximation and projection (UMAP) analysis, we identified eight distinct T$_{reg}$ cell clusters based on the expression of well-characterized markers, with a particular focus on two clusters of activated T$_{reg}$ cells that exhibited higher expression of markers and gene sets relative to naive T$_{reg}$ cells (*Figure 5A*, *Figure 5—figure supplement 1A–C*). A comparison between *Cxxc1*-deficient and WT T$_{reg}$ cells within each cluster revealed a reduction in *Cxxc1*-deficient cells in the naive subsets, while an increase was observed in the Gzmb[+] and H2-Eb1[+] subsets (*Figure 5B*). To further elucidate the transition of T$_{reg}$ cells along a dynamic biological timeline, we constructed pseudo-time trajectories using Slingshot (*Street et al., 2018*). The pseudo-time gradient depicted a progression from quiescent to activated T$_{reg}$ cells, ultimately encompassing the Gzmb[+] and H2-Eb1[+] subsets (*Figure 5C*). Given the antigen-specific suppression capabilities of T$_{reg}$ cells (*Hori et al., 2002*; *Tarbell et al., 2004*), we examined their clonal expansion. The analysis revealed that expanded WT TCR clonotypes (*n*≥2) were predominantly distributed among the Nt5e[+] subsets, while *Cxxc1*-deficient T$_{reg}$ cells showed expanded clonotypes primarily within the Gzmb[+] and H2-Eb1[+] subsets (*Figure 5D*, *Figure 5—figure supplement 1D*). TCR sharing analysis indicated clonotype sharing among various clusters of WT T$_{reg}$ cells, suggesting a degree of homogeneity. However, the reduced TCR sharing in *Cxxc1*-deficient T$_{reg}$ cells implies that decreased TCR diversity may impair the suppressive activity of T$_{reg}$ cells (*Figure 5E*; *Dikiy and Rudensky, 2023*). Furthermore, the *Cxxc1*-deficient group exhibited lower expression of several T$_{reg}$-specific genes associated with suppressive functions, such as *Nt5e*, *Il10*, *Pdcd1*, *Klrg1*, as well as genes that inhibit effector T cell differentiation, including *Sell* and *Tcf7* (*Figure 5F*, *Figure 5—figure supplement 1E*). Conversely, *Cxxc1*-deficient T$_{reg}$ cells demonstrated elevated expression of *Gzmb*, *Il2ra*, and *Cd69* compared to WT, reflecting a profile indicative of increased activation (*Figure 5F*,

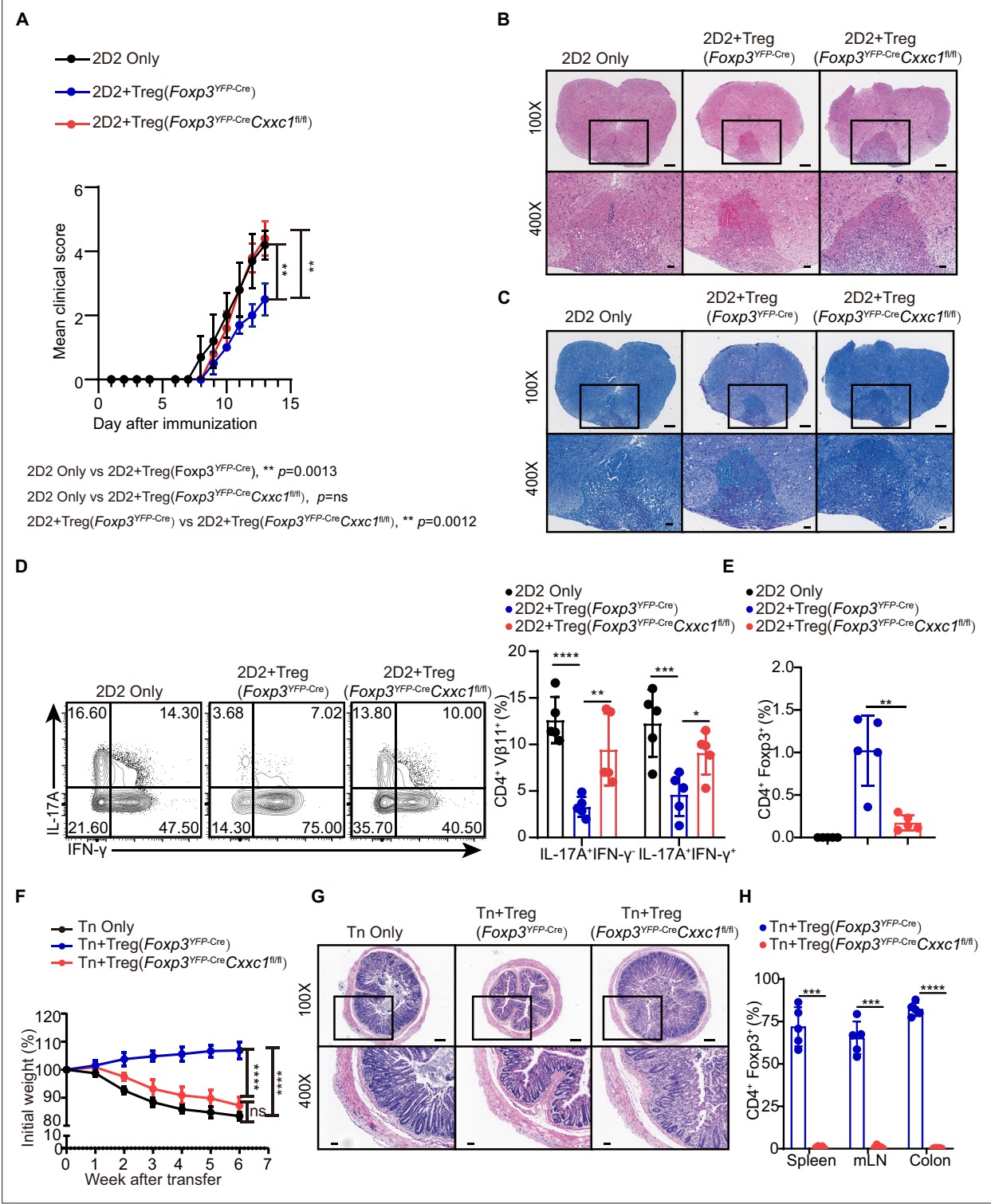

**Figure 4.** CXXC1 is essential for T$_{reg}$ cells to suppress T cell-mediated experimental autoimmune encephalomyelitis (EAE) and colitis. (**A**) Mean clinical scores for EAE in *Rag1*$^{-/-}$ recipients of 2D2 CD4$^+$ T cells, either alone or in combination with *Foxp3*$^{YFP-Cre}$ or *Foxp3*$^{YFP-Cre}$*Cxxc1*$^{fl/fl}$ mice after immunization with MOG35–55, complete Freund's adjuvant (CFA), and pertussis toxin (*n* = 5). (**B, C**) Representative histology of the spinal cord of *Rag1*$^{-/-}$ mice after EAE induction. Hematoxylin and eosin (H&E) staining (upper), Luxol fast blue (F&B) staining (lower). Scale bars, 50 μm (×400) and 200 μm (×100). (**D**) Representative flow cytometry plots and quantification of the percentages of IFNγ$^+$ or IL-17A$^+$ CD4$^+$Vβ11$^+$ T cells (*n* = 5). (**E**) Statistical analysis of the percentage CD4$^+$ FOXP3$^+$ T$_{reg}$ cell in central nervous system (CNS) tissues 14 days after EAE induction (*n* = 5). (**F**) Changes in body weight of *Rag1*$^{-/-}$ mice after colitis induction (*n* = 6). (**G**) H&E staining of colons from T cell-induced colitis mice 6 weeks after T cell transfer. Scale bars, 50 μm (×400) and

*Figure 4 continued on next page*

Figure 4 continued

200 μm (×100). (**H**) Statistical analysis of the percentage CD4$^+$ FOXP3$^+$ T$_{reg}$ cell in the spleen, mesenteric lymph nodes, and colon 6 weeks after colitis induction (n = 5). Error bars show mean ± SD. p values are determined by a unpaired t-test or two-way ANOVA and Holm–Sidak post hoc test (**A, D, E, F, H**) (*p < 0.05, **p < 0.01, ***p < 0.001, ****p < 0.0001).

The online version of this article includes the following source data and figure supplement(s) for figure 4:

**Source data 1.** Original source data for graphs displayed in *Figure 4*.

**Figure supplement 1.** The number of T$_{reg}$ cells deficient in CXXC1 did not decrease.

**Figure supplement 1—source data 1.** Original source data for graphs displayed in *Figure 4—figure supplement 1*.

***G***). Additionally, we observed increased expression of genes linked to Th1-type inflammation, such as *Ifng*, *Tbx21*, and *Hif1a*, in *Cxxc1*-deficient T$_{reg}$ cells, likely due to extreme inflammatory conditions (***Figure 5F–H***). The proportion of FOXP3$^+$Ki67$^+$ T$_{reg}$ cells was lower in cKO mice compared to WT mice (***Figure 5I***). These findings underscore the crucial role of CXXC1 in maintaining T$_{reg}$ cell function and homeostasis.

## Intrinsic *Cxxc1* deficiency impairs T$_{reg}$ cell suppression function, proliferation, and molecular programs

To confirm that the deficiency in T$_{reg}$ cell function in T$_{reg}$-specific *Cxxc1*-deficient animals is due to intrinsic defects caused by *Cxxc1* deficiency, rather than severe autoimmune inflammation in *Foxp3*$^{YFP-Cre}$*Cxxc1*$^{fl/fl}$ mice, we examined *Cxxc1*-sufficient and *Cxxc1*-deficient T$_{reg}$ cell subsets in heterozygous *Foxp3*$^{YFP-Cre/+}$ *Cxxc1*$^{fl/fl}$ (designated as 'het-KO') and littermate *Foxp3*$^{YFP-Cre/+}$ *Cxxc1*$^{fl/+}$ (designated as 'het-WT') female mice (***Figure 6A***). Notably, het-KO female mice did not exhibit overt signs of autoimmunity, as random X-chromosome inactivation led to the coexistence of both *Cxxc1*-cKO and *Cxxc1*-WT T$_{reg}$ cells. However, both the frequency and absolute numbers of FOXP3$^+$YFP$^+$ T$_{reg}$ cells within the total T$_{reg}$ population were reduced in het-KO mice compared to their counterparts in het-WT littermates, indicating that *Cxxc1* deficiency imposes a competitive disadvantage on T$_{reg}$ cells (***Figure 6B***). Additionally, *Cxxc1*-deficient YFP$^+$ T$_{reg}$ cells failed to upregulate the proliferation marker Ki-67 (***Figure 6C***). Moreover, YFP$^+$ T$_{reg}$ cells in het-KO female mice showed reduced expression of key genes essential for suppressive function, including ICOS, CD25, CTLA4, and GITR, compared to YFP$^-$T$_{reg}$ cells from the same mice (***Figure 6D***). Consistently, we confirmed the impaired suppressive function of T$_{reg}$ cells from heterozygous *Foxp3*$^{YFP-Cre/+}$ *Cxxc1*$^{fl/fl}$ mice in vitro and in vivo (***Figure 6E***, ***Figure 6—figure supplement 1A–E***). To investigate the molecular program affected by the deletion of *Cxxc1* in T$_{reg}$ cells, we performed RNA-seq analysis on CD4$^+$YFP$^+$ T$_{reg}$ cells isolated from het-WT and het-KO mice. We then conducted a differential gene expression (DGE) analysis based on the RNA-seq data. Among all expressed genes, 865 were upregulated and 761 were downregulated in CD4$^+$YFP$^+$ T$_{reg}$ cells from het-KO mice compared to het-WT mice (***Figure 6—figure supplement 1F***). GO enrichment analysis revealed that the downregulated genes in *Cxxc1*-deficient T$_{reg}$ cells were predominantly enriched in pathways related to the negative regulation of immune system process and regulation of cell–cell adhesion (***Figure 6—figure supplement 1G***). The *Cxxc1*-deficient T$_{reg}$ cells also showed reduced expression of several genes associated with T$_{reg}$ cell suppressive function, including *Il10*, *Tigit*, *Lag3*, *Icos*, *Nt5e*(encoding CD73) and *Itgae* (encoding CD103) (***Figure 6—figure supplement 1H***). Thus, while YFP$^-$ WT T$_{reg}$ cells effectively prevent autoimmunity in het-KO mice, the absence of *Cxxc1* in YFP$^+$ T$_{reg}$ cells disrupts key T$_{reg}$ cell marker expression and impairs their suppressive function under steady-state conditions.

## The FOXP3–CXXC1 complex regulates the expression of key factors in T$_{reg}$ cells that are associated with the breadth of H3K4me3

CXXC1 binds to unmethylated CpG DNA via its N-terminal CXXC-finger domain, facilitating its interaction with DNA methyltransferase 1 (DNMT1). This binding stabilizes the DNMT1 protein, thereby regulating DNA methylation (***Butler et al., 2008***; ***Butler et al., 2009***). To investigate whether CXXC1 depletion affects DNA methylation in T$_{reg}$ cells, we performed whole genome bisulfite sequencing (WGBS) on T$_{reg}$ cells isolated from both WT and cKO mice. On average, *Cxxc1*-deficient T$_{reg}$ cells exhibited no changes in DNA methylation at gene loci or across genome-wide CpG sites, irrespective of chromosomal region (***Figure 7—figure supplement 1A–C***). Furthermore, *Cxxc1* knockout T$_{reg}$ cells

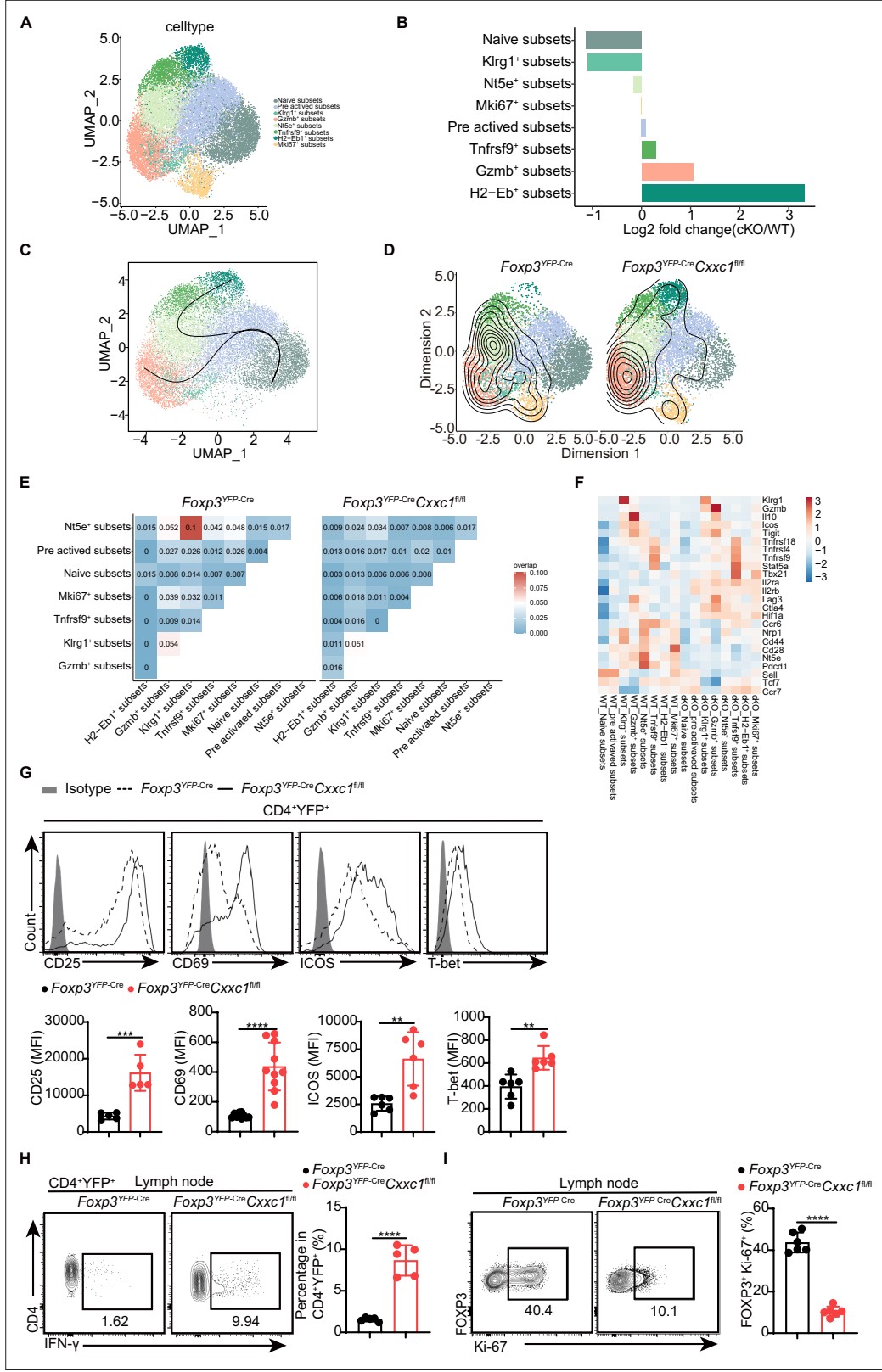

**Figure 5.** Single-cell transcriptomics reveals distinct T$_{reg}$ cell populations. (**A**) Uniform manifold approximation and projection (UMAP) plot showing clusters identified based on variable gene expression of sorted CD4+YFP+ T$_{reg}$ cells. Each dot represents a cell, and each color corresponds to a different population of cell types. Clustering analysis revealed eight distinct T$_{reg}$ cell populations. (**B**) Mean fold changes in cluster abundance between

*Figure 5 continued on next page*

*Figure 5 continued*

*Foxp3*[YFP-Cre] and *Foxp3*[YFP-Cre]*Cxxc1*[fl/fl] mice. (**C**) Pseudotime trajectories of T$_{reg}$ cells based on Slingshot, color-coded by T$_{reg}$ cell subpopulations. (**D**) Visualization of density and clonotype richness across T$_{reg}$ clusters from *Foxp3*[YFP-Cre] and *Foxp3*[YFP-Cre]*Cxxc1*[fl/fl] mice. (**E**) T cell receptor (TCR) sharing of expanded clonotypes across all possible combinations of T$_{reg}$ cells from *Foxp3*[YFP-Cre] and *Foxp3*[YFP-Cre]*Cxxc1*[fl/fl] mice. (**F**) Heatmap showing *Z* scores for the average expression of T$_{reg}$-specific genes in each cluster between *Foxp3*[YFP-Cre] and *Foxp3*[YFP-Cre]*Cxxc1*[fl/fl] mice. Representative flow cytometry plots and quantification of (**G**) expression of CD25, CD69, ICOS, T-bet, and (**H**) IFN-γ in CD4$^+$YFP$^+$ T$_{reg}$ cells from *Foxp3*[YFP-Cre] and *Foxp3*[YFP-Cre]*Cxxc1*[fl/fl] mice (*n* = 5 CD25, *n* = 10 CD69, *n* = 5 ICOS, *n* = 6 T-bet, *n* = 5 IFN-γ). (**I**) Ki-67 expression (left) and frequency (right) in CD4$^+$FOXP3$^+$ T$_{reg}$ cells from *Foxp3*[YFP-Cre] and *Foxp3*[YFP-Cre]*Cxxc1*[fl/fl] mice (*n* = 6). Error bars show mean ± SD. p values are determined by a unpaired *t*-test (**G–I**) (**p < 0.01, ***p < 0.001, ****p < 0.0001). The flow cytometry results are representative of three independent experiments.

The online version of this article includes the following source data and figure supplement(s) for figure 5:

**Source data 1.** Original source data for graphs displayed in *Figure 5*.

**Source data 2.** Single-cell T cell receptor (TCR) V(D)J repertoire profiling of CD4$^+$YFP $^+$ T$_{reg}$ cells in *Foxp3*[YFP-Cre] and *Foxp3*[YFP-Cre]*Cxxc1*[fl/fl] mice.

**Figure supplement 1.** Single-cell transcriptomics reveals distinct T$_{reg}$ cell populations.

**Figure supplement 1—source data 1.** Original source data for graphs displayed in *Figure 5—figure supplement 1*.

did not show an increase in DNA methylation at key T$_{reg}$ signature gene loci (*Figure 7—figure supplement 1D*). Given the pivotal role of MLL4-mediated H3K4me1 in establishing the enhancer landscape and facilitating long-range chromatin interactions during T$_{reg}$ cell development (*Placek et al., 2017*), we performed CUT&Tag to assess changes in H3K4me1 levels in *Cxxc1*-deficient T$_{reg}$ cells. This analysis revealed that H3K4me1 levels were similar in both WT and *Cxxc1*-deficient T$_{reg}$ cells (*Figure 7—figure supplement 1E–G*).

While H3K4me3 modifications typically form sharp 1- to 2-kb peaks around promoters, some genes exhibit broader H3K4me3 regions, referred to as broad H3K4me3 domains (H3K4me3-BDs), which can extend to cover part or all of the gene's coding sequences (up to 20 kb) (*Benayoun et al., 2014*; *Zacarías-Cabeza et al., 2015*). Broad H3K4me3 domains are preferentially associated with genes essential for the identity or function of specific cell types (*Benayoun et al., 2014*; *Chen et al., 2015*) and have been implicated in enhancing transcriptional elongation and increasing enhancer activity (*Chen et al., 2015*). To further explore the relationship between broad H3K4me3 domains and the expression of immune-regulatory genes, we analyzed genes enriched with broad H3K4me3 regions. We classified the H3K4me3 domains surrounding TSSs into three categories: broad (more than 5 kb), medium (between 1 and 5 kb), and narrow (less than 1 kb) (*Figure 7A*). Notably, *Cxxc1*-deficient T$_{reg}$ cells exhibited weaker H3K4me3 signals compared to WT cells within the broad H3K4me3 domains where CXXC1 binding is prominent (*Figure 7A, B*). Using the criteria established by *Benayoun et al., 2014*, which defines the top 5% of the widest H3K4me3 domains as BDs, we observed similar enrichment results (*Figure 7—figure supplement 1H, I*). We then compared three groups of genes: BD-associated genes with reduced H3K4me3 levels following *Cxxc1* deletion, genes with direct CXXC1 binding, and genes with direct FOXP3 binding. The Venn diagram revealed that the majority of genes (283 out of 294, 96%) with CXXC1 binding and reduced H3K4me3 levels overlap with FOXP3-bound genes, suggesting that CXXC1 is functionally associated with FOXP3-regulated loci within broad H3K4me3 domains (*Figure 7C*). Furthermore, GO term analysis indicated that BD-associated genes are enriched in biological processes related to the negative regulation of immune system processes (*Figure 7D*). Genome browser views displayed the enrichments of FOXP3, CXXC1, and H3K4me3 at key signature genes in T$_{reg}$ cells, such as *Ctla4, Il2ra, Icos*, and *Tnfrsf18*, with lower H3K4me3 densities observed at these loci in *Cxxc1*-deficient T$_{reg}$ cells (*Figure 7E*). Similar patterns were observed at core genes involved in T$_{reg}$ homeostasis and suppressive function (e.g., *Lag3, Nt5e, Ikzf4*, and *Cd28*) (*Figure 7—figure supplement 1J*; *Gokhale et al., 2019*; *Zhang et al., 2013*). These findings suggest that CXXC1 and FOXP3 collaboratively promote sustained T$_{reg}$ cell homeostasis and function by preserving the H3K4me3 modification at key T$_{reg}$ cell genes.

We previously demonstrated that the FOXP3–CXXC1 complex plays a key role in modulating H3K4me3 deposition at T$_{reg}$-specific gene loci. To further clarify whether *Cxxc1* deletion affects FOXP3

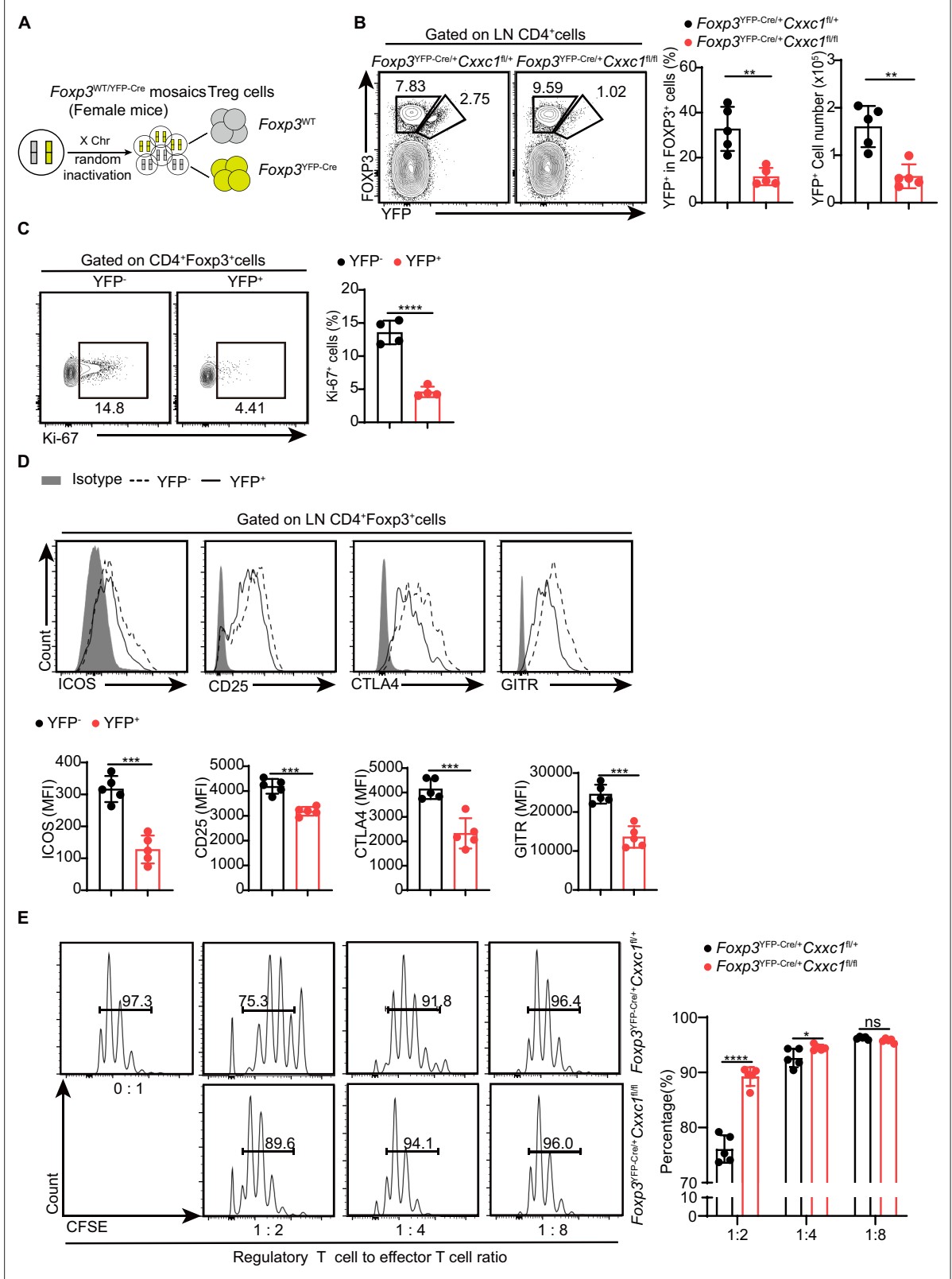

**Figure 6.** *Cxxc1*-deficient T$_{reg}$ cells exhibit functional impairment and disrupted homeostasis in steady-state conditions. (**A**) Schematic representation of wild-type and Cre-positive T$_{reg}$ cells in female *Foxp3*$^{YFP-Cre/+}$ mice.(**B**) Flow cytometry analysis of the YFP$^-$FOXP3$^+$ (WT) and YFP$^+$FOXP3$^+$ (KO) T$_{reg}$ cells in *Foxp3*$^{YFP-Cre/+}$ *Cxxc1*$^{fl/+}$ (het-WT) and *Foxp3*$^{YFP-Cre /+}$ *Cxxc1*$^{fl/fl}$ (het-KO) female mice (left), along with the frequency and absolute numbers of YFP$^+$ cells within the total T$_{reg}$ population (right) (*n* = 5). (**C**) Flow cytometry analysis of Ki-67expression (left) and MFI (right) in YFP$^-$ and YFP$^+$ cells within the CD4$^+$FOXP3$^+$

*Figure 6 continued on next page*

*Figure 6 continued*

T$_{reg}$ cells from 6- to 8-week-old het-KO female mice (*n* = 4). (**D**) Representative flow cytometry plots and quantification of ICOS, CD25, CTLA4, and GITR expression in YFP$^-$ and YFP$^+$ cells within CD4$^+$FOXP3$^+$ T$_{reg}$ cells from het-KO female mice (*n* = 5). (**E**) Suppression of CFSE-labeled Tn cell proliferation by different ratios of CD4$^+$YFP$^+$ T$_{reg}$ cells from *Foxp3*$^{YFP-Cre/+}$ *Cxxc1*$^{fl/+}$ and *Foxp3*$^{YFP-Cre/+}$ *Cxxc1*$^{fl/fl}$ female mice. On the right, the percentage of proliferated responding T cells is presented (*n* = 5). Error bars show mean ± SD. p values are determined by a unpaired *t*-test or multiple unpaired *t*-test (**B–E**) (ns, not significant. *p < 0.05, **p < 0.01, ***p < 0.001, ****p < 0.0001). The flow cytometry results are representative of three independent experiments.

The online version of this article includes the following source data and figure supplement(s) for figure 6:

**Source data 1.** Original source data for graphs displayed in *Figure 6*.

**Figure supplement 1.** *Cxxc1* deficiency impairs T$_{reg}$ cell suppressive function and aggravates experimental autoimmune encephalomyelitis (EAE) severity.

**Figure supplement 1—source data 1.** Original source data for graphs displayed in *Figure 6—figure supplement 1*.

binding to its target genes, we performed CUT&Tag experiments to compare FOXP3-binding profiles between WT and *Cxxc1* KO T$_{reg}$ cells. The results revealed that most FOXP3-bound regions in WT T$_{reg}$ cells were similarly enriched in KO T$_{reg}$ cells, indicating that *Cxxc1* deletion does not impair FOXP3's DNA-binding ability (*Figure 7—figure supplement 2A, B*). Together, these findings suggest that the regulatory role of CXXC1 in T$_{reg}$ cells is mediated through its effect on H3K4me3 deposition rather than altering FOXP3's binding to DNA.

## Discussion

T$_{reg}$ cells specifically express the transcription factor FOXP3, which is essential for maintaining T$_{reg}$ lineage stability and suppressive function (*Wan and Flavell, 2007*). However, FOXP3 alone is insufficient to fully regulate the transcriptional signature and functionality of T$_{reg}$ cells; its interaction with protein partners is crucial for this regulation. In this study, we identify CXXC1 as a critical epigenetic regulator and functional cofactor of FOXP3. Although CXXC1 is not required for FOXP3's DNA-binding activity, as evidenced by similar FOXP3-binding patterns in WT and *Cxxc1*-deficient T$_{reg}$ cells (CUT&Tag analysis), it plays an essential role in maintaining H3K4me3 modifications at FOXP3 target loci. By acting as both an epigenetic regulator and a FOXP3 cofactor, CXXC1 ensures the stability of the T$_{reg}$ transcriptional program, highlighting its pivotal role in preserving T$_{reg}$ cell functionality and immune homeostasis.

FOXP3 is known to promote histone H3 acetylation at the promoters and enhancers of its target genes, such as *Il2ra*, *Ctla4*, and *Tnfrsf18*, following T$_{reg}$ cell activation, thereby functioning as a transcriptional activator (*Chen et al., 2006*; *Morikawa et al., 2014*). Conversely, FOXP3 can act as a repressor by silencing target genes like *Il2* and *Ifng* through the induction of histone H3 deacetylation, mediated by the recruitment of histone deacetylases and transcriptional co-repressors (*Pan et al., 2009*; *Wang et al., 2015*). Additionally, FOXP3 exerts this repression by recruiting the Ezh2-containing polycomb repressive complex to target genes during activation, as FOXP3-repressed genes are associated with H3K27me3 deposition and reduced chromatin accessibility (*Arvey et al., 2014*; *DuPage et al., 2015*). In parallel, recent studies have begun to explore the biological significance of H3K4me3 breadth, revealing a positive correlation between H3K4me3 breadth and gene expression (*Chen et al., 2015*; *Liu et al., 2016*; *Sze et al., 2020*). These studies also suggest that H3K4me3 breadth contributes to defining specific cell identities during development and disease, including systemic autoimmune diseases like systemic lupus erythematosus and various cancers (*Benayoun et al., 2014*; *Dahl et al., 2016*; *Wong et al., 2015*; *Zhang et al., 2016b*; *Zhang et al., 2016a*). Our study demonstrates that the loss of *Cxxc1* leads to reduced H3K4me3 levels, predominantly in genes with broader peaks, such as *Il2ra*, *Tnfrsf18*, *Ctla4*, and *Icos*, directly impairing their suppressive function in T$_{reg}$ cells.

The T$_{reg}$-specific deletion of *Cxxc1* leads to a rapid and fatal autoimmune disorder, characterized by systemic inflammation and tissue damage, underscoring the essential role of CXXC1 in maintaining immune self-tolerance within T$_{reg}$ cells. Interestingly, despite this severe phenotype, our findings show that H3K4me1 levels were comparable between WT and *Cxxc1*-deficient T$_{reg}$ cells. In contrast, MLL4 is critical for T$_{reg}$ cell development in the thymus, primarily by regulating H3K4me1, though it is not required for peripheral T$_{reg}$ cell function (*Placek et al., 2017*). Numerous studies investigating the effectors of T$_{reg}$ cell-mediated suppression have identified a wide range of molecules and mechanisms. These include the upregulation of the inhibitory co-stimulatory receptor CTLA-4, which initiates

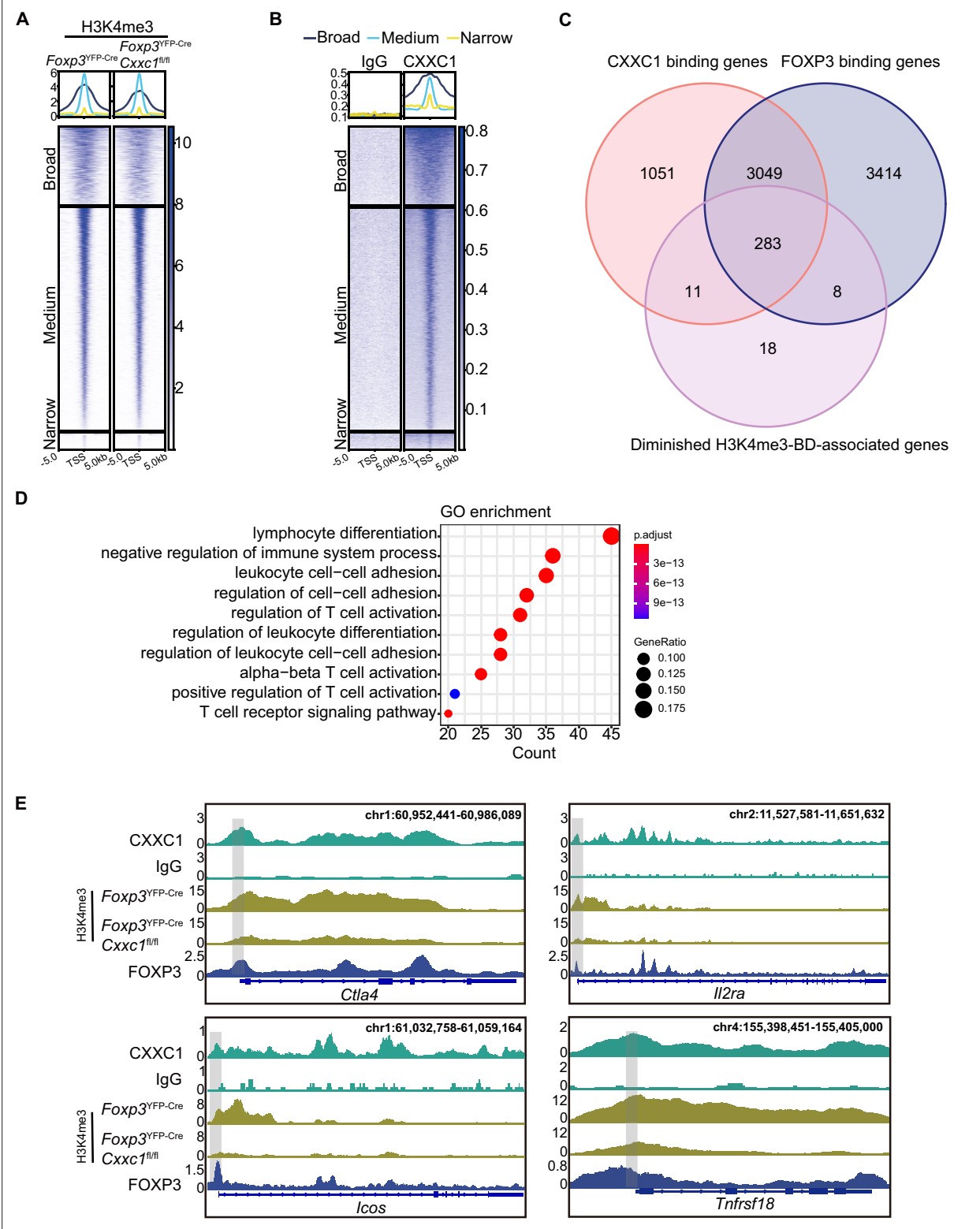

**Figure 7.** CXXC1 regulates FOXP3-dependent molecule H3K4 trimethylation in T$_{reg}$ cells. Heatmaps showing H3K4me3 (**A**) and CXXC1 (**B**) signals centered on narrow, medium, and broad domains. The top panel shows the average CUT&Tag signals around indicated domains. (**C**) Venn diagrams showing the overlap of FOXP3-binding genes, CXXC1-binding genes, and H3K4me3-BD-associated genes with decreased H3K4me3 levels after *Cxxc1* depletion in T$_{reg}$ cells. (**D**) Gene Ontology (GO) pathway analysis of the overlapped genes in **C**. (**E**) Representative genome browser view showing the enrichments of FOXP3, CXXC1, and H3K4me3 in T$_{reg}$ cells.

*Figure 7 continued on next page*

*Figure 7 continued*

The online version of this article includes the following source data and figure supplement(s) for figure 7:

**Source data 1.** Table: list of overlapped genes in *Figure 7C*.

**Figure supplement 1.** The epigenetic program of T$_{reg}$ cells.

**Figure supplement 2.** FOXP3-binding profile and enrichment in WT and *Cxxc1*-Null T$_{reg}$ cells.

inhibitory signaling; the sequestration of the T cell growth factor IL-2 via CD25; the secretion of inhibitory cytokines, such as interleukin (IL)-10, IL-35, and TGF-β and the activity of ectoenzymes CD39 and CD73 on the T$_{reg}$ cells surface, which convert extracellular ATP, a potent pro-inflammatory mediator, into its anti-inflammatory counterpart, adenosine (*Deaglio et al., 2007*; *Dikiy and Rudensky, 2023*). Mechanistically, we demonstrated that CXXC1 interacts with FOXP3 to regulate T$_{reg}$ cell function by trimethylating H3K4 at broad H3K4me3 domains of multiple genes involved in suppressive functions such as *Il2ra, Nt5e, and Ctla4*. Additionally, MLL1, another KMT, controls T$_{reg}$ cell activation and function by specifically regulating H3K4 trimethylation at genes encoding key T$_{reg}$-related molecules such as *Tigit*, *Klrg1*, *Tbx21*, *Cxcr3*, and serves as a crucial epigenetic regulator in establishing a stable Th1-T$_{reg}$ lineage (*Wang et al., 2024*).

Our findings demonstrate that *Cxxc1*-deficient T$_{reg}$ cells exhibit reduced H3K4me3 levels at T$_{reg}$-specific loci, indicating that CXXC1 plays a critical role in regulating this epigenetic modification. This is consistent with previous studies showing that CXXC1 acts as a non-catalytic component of the Set1/COMPASS complex (*Brown et al., 2017*; *Lee and Skalnik, 2005*; *Shilatifard, 2012*; *Thomson et al., 2010*), which includes the H3K4 methyltransferases SETD1A and SETD1B, the primary enzymes responsible for H3K4 trimethylation. Based on these insights, we propose that CXXC1 supports H3K4me3 deposition in T$_{reg}$ cells by interacting with and stabilizing the activity of the Set1/COMPASS complex. Further studies are required to directly investigate the interactions between CXXC1 and these methyltransferases in the T$_{reg}$ cell context.

Our findings provide novel insights into the suppressive functions, heterogeneity, and regulatory mechanisms of T$_{reg}$ cells. Maintaining T$_{reg}$ cell homeostasis and function remains a challenge in harnessing T$_{reg}$ cells for the treatment of autoimmune diseases and the prevention of graft rejection.

## Materials and methods

**Key resources table**

| Reagent type (species) or resource | Designation | Source or reference | Identifiers | Additional information |
|---|---|---|---|---|
| Strain, strain background (*Mus musculus*) | *Foxp3*$^{YFP-Cre}$ mice | Gifted from Prof. Bin Li | Shanghai Jiao Tong University | N/A |
| Strain, strain background (*Mus musculus*) | *Cxxc1*$^{fl/fl}$ mice | The Shanghai Research Center for Model Organisms | N/A | N/A |
| Strain, strain background (*Mus musculus*) | C57BL/6JGpt-Rag1$^{em1Cd3259}$/Gpt, *Rag1*$^{-/-}$ mice | GemPharmatech | Cat# T004753; RRID:IMSR_GPT:T004753 | N/A |
| Strain, strain background (*Mus musculus*) | 2D2 mice | Gifted from Prof. Linrong Lu | Zhejiang University | N/A |
| Biological sample (*Mus musculus*) | Thymus, lymph node, lung, liver, small intestine lamina propria lymphocytes | This paper | N/A | Freshly isolated tissue |
| Cell line (*Mus musculus*, mouse) | HEK 293T | ATCC | Cat# ACS-4500; RRID:CVCL_4V93 | N/A |
| Cell line (*Mus musculus*, mouse) | Plat E | Gifted from Prof. Xiaolong Liu | Shanghai Institutes for Biological Sciences | N/A |
| Antibody | anti-mouse CD16/32 (rat monoclonal) | BioLegend | Cat# 101320; RRID:AB_1574975 | (1:200) |

*Continued on next page*

*Continued*

| Reagent type (species) or resource | Designation | Source or reference | Identifiers | Additional information |
|---|---|---|---|---|
| Antibody | PE/Cyanine7 anti-TCR-β (hamster monoclonal) | BioLegend | Cat# 109222; RRID:AB_893625 | (1:400) |
| Antibody | PE/Cyanine7 anti-KLRG1(hamster monoclonal) | BioLegend | Cat# 138416; RRID:AB_2561736 | (1:100) |
| Antibody | APC-eFluo 780 anti-CD4 (rat monoclonal) | Invitrogen | Cat# 47-0042-82; RRID:AB_1272183 | (1:400) |
| Antibody | PE anti-CD152 (CTLA4) (hamster monoclonal) | BioLegend | Cat# 106306; RRID:AB_313255 | (1:100) |
| Antibody | PE anti-CD278 (ICOS) (hamster monoclonal) | BioLegend | Cat# 107706; RRID:AB_313335 | (1:100) |
| Antibody | PE anti-CD357 (GITR) (rat monoclonal) | BioLegend | Cat# 126310; RRID:AB_1089132 | (1:100) |
| Antibody | PE anti-CD69 (hamster monoclonal) | Invitrogen | Cat# 12-0691-83; RRID:AB_ 465733 | (1:400) |
| Antibody | PE anti-CD25 (rat monoclonal) | BioLegend | Cat# 102008; RRID:AB_312856 | (1:400) |
| Antibody | Brilliant Violet 650 anti-mouse CD8a (rat monoclonal) | BioLegend | Cat# 100742; RRID:AB_2563056 | (1:400) |
| Antibody | APC anti-TCR V beta 11 (rat monoclonal) | Invitrogen | Cat# 17-5827-82; RRID:AB_2573226 | (1:400) |
| Antibody | APC/Cyanine7 anti-CD44 (rat monoclonal) | BioLegend | Cat# 103028; RRID:AB_830785 | (1:400) |
| Antibody | APC anti-CD62L (rat monoclonal) | BioLegend | Cat# 104412; RRID:AB_313099 | (1:400) |
| Antibody | APC anti-Foxp3 (rat monoclonal) | Invitrogen | Cat# 17-5773-82; RRID:AB_469457 | (1:100) |
| Antibody | Pacific Blue anti-IFN-γ (rat monoclonal) | BioLegend | Cat# 505818; RRID:AB_893526 | (1:100) |
| Antibody | PE anti-IL-4 (rat monoclonal) | Invitrogen | Cat# 12-7041-82; RRID:AB_466156 | (1:100) |
| Antibody | PE-Cyanine7 anti-IL-17A (rat monoclonal) | Invitrogen | Cat# 25-7177-82; RRID:AB_10732356 | (1:100) |
| Antibody | PE anti- T-bet (mouse monoclonal) | Invitrogen | Cat# 12-5825-82; RRID:AB_925761 | (1:100) |
| Antibody | BV421 anti-PD-1 (hamster monoclonal) | BD | Cat# 562584; RRID:AB_2737668 | (1:100) |
| Antibody | APC anti-CD45RB (rat monoclonal) | BioLegend | Cat# 103319; RRID:AB_2565228 | (1:100) |
| Antibody | PE anti-CD73 (rat monoclonal) | BioLegend | Cat# 127205; RRID:AB_ 2154094 | (1:100) |
| Antibody | PerCPCy5.5 anti-Ki-67(mouse monoclonal) | BD | Cat# 561284; RRID:AB_10611574 | (1:200) |
| Antibody | Anti-Cxxc1 (Rabbit monoclonal) | Abcam | Cat# ab198977 RRID:AB_3101764 | WB (1:1000), IF (1:100) |
| Antibody | Anti-Foxp3 (mouse monoclonal) | Invitrogen | Cat# 14-4774-82; RRID:AB_467552 | WB (1:1000) |
| Antibody | Anti-Flag (Rabbit monoclonal) | Cell Signaling | Cat# 14793; RRID:AB_2572291 | WB (1:1000) |

*Continued on next page*

*Continued*

| Reagent type (species) or resource | Designation | Source or reference | Identifiers | Additional information |
|---|---|---|---|---|
| Antibody | Anti-HA (Rabbit monoclonal) | Cell Signaling | Cat# 3724S RRID:AB_1549585 | WB (1:1000) |
| Antibody | anti-H3K4me3 (Rabbit Polyclonal) | Active Motif | Cat# 39016 RRID:AB_2687512 | CUT&Tag (1:50) |
| Antibody | anti-FOXP3 (Rabbit Polyclonal) | Abcam | Cat# ab150743 | CUT&Tag (1:50) |
| Antibody | Normal Rabbit IgG (Rabbit Polyclonal) | Cell Signaling | Cat# 2729 RRID:AB_1031062 | CUT&Tag (1:50) |
| Antibody | HRP-conjugated anti-mouse IgG | SouthernBiotech | Cat# 1033-05 RRID:AB_2737432 | (1:2000) |
| Antibody | HRP-conjugated anti-mouse IgE | SouthernBiotech | Cat# 1110-05 RRID:AB_2794604 | (1:2000) |
| Antibody | Anti-Mo CD3e (hamster monoclonal) | Invitrogen | Cat# 16-0031-85; RRID:AB_468848 | 2 µg/ml |
| Antibody | Anti-Mo CD28 (hamster monoclonal) | Invitrogen | Cat# 16-0281-85; RRID:AB_468922 | 3 µg/ml |
| Antibody | Anti-mouse IFN-γ (rat monoclonal) | BioLegend | Cat# 505847; RRID:AB_2616675 | 10 µg/ml |
| Antibody | Anti-mouse IL-12 (rat monoclonal) | BioLegend | Cat# 505309; RRID:AB_2783330 | 10 µg/ml |
| Antibody | Anti-mouse IL-4 (rat monoclonal) | BioLegend | Cat# 504135; RRID:AB_2750404 | 10 µg/ml |
| Sequence-based reagent | *Cxxc1* genotyping Forward | This paper | Genotyping PCR primer | CGAGAGATGAAGAGGAGCCA |
| Sequence-based reagent | *Cxxc1* genotyping Reverse | This paper | Genotyping PCR primer | CACAAAGATAGGCTCCATCC |
| Sequence-based reagent | *Foxp3*$^{YFP\text{-}Cre}$ WT genotyping Forward | This paper | Genotyping PCR primer | CTATGGAAACCGGGCGATGA |
| Sequence-based reagent | *Foxp3*$^{YFP\text{-}Cre}$ WT genotyping Reverse | This paper | Genotyping PCR primer | AGTGGCAAGTGAGACGTGGG |
| Sequence-based reagent | *Foxp3*$^{YFP\text{-}Cre}$ genotyping Forward | This paper | Genotyping PCR primer | AGGATGTGAGGGACTACCTCCTGTA |
| Sequence-based reagent | *Foxp3*$^{YFP\text{-}Cre}$ genotyping Reverse | This paper | Genotyping PCR primer | TCCTTCACTCTGATTCTGGCAATTT |
| Sequence-based reagent | *Actb* qPCR Forward | This paper | qRT-PCR primer | CTGTCCCTGTATGCCTCTG |
| Sequence-based reagent | *Actb* qPCR Reverse | This paper | qRT-PCR primer | ATGTCACGCACGATTTCC |
| Sequence-based reagent | *Cxxc1* qPCR Forward | This paper | qRT-PCR primer | CTGTGGAGAAGATTTGTGGG |
| Sequence-based reagent | *Cxxc1* qPCR Reverse | This paper | qRT-PCR primer | TCTTGTTGTCTAGAGTGGCGATCT |
| Recombinant DNA reagent | pCMV-C-HA plasmid | This paper | N/A | N/A |
| Recombinant DNA reagent | p3×Flag-CMV7.1 plasmid | This paper | N/A | N/A |
| Commercial assay or kit | MojoSort Mouse CD4 T Cell Isolation Kit | BioLegend | Cat# 480005 | N/A |
| Commercial assay or kit | MojoSort Mouse CD4 Naive T Cell Isolation Kit | BioLegend | Cat# 480039 | N/A |
| Commercial assay or kit | RNeasy Plus Mini Kit | QIAGEN | Cat# 74134 | N/A |

*Continued on next page*

*Continued*

| Reagent type (species) or resource | Designation | Source or reference | Identifiers | Additional information |
|---|---|---|---|---|
| Commercial assay or kit | TruePrep DNA Library Prep Kit V2 for Illumina | Vazyme | Cat# TD501 | N/A |
| Commercial assay or kit | Hyperactive In-Situ ChiP Library Prep Kit for Illumina | Vazyme | Cat# TD901 | N/A |
| Commercial assay or kit | ClonExpress II One Step Cloning Kit | Vazyme | Cat# C112-01 | N/A |
| Commercial assay or kit | E.Z.N.A. Endo-free Plasmid Mini Kit II | Omega | Cat# D6950-02 | N/A |
| Commercial assay or kit | BD Mouse Immune Single-Cell Multiplexing Kit | BD | Cat# 633793 | N/A |
| Commercial assay or kit | BD Mouse Immune Single-Cell Multiplexing Kit | BD | Cat# 633801 | N/A |
| Recombinant protein | IL-2 | Peprotech | Cat# AF-212-12-20ug | 50 U/ml |
| Recombinant protein | TGF-β | Peprotech | Cat# 100-21C-250ug | 5 ng/ml |
| Chemical compound, drug | PMA | Sigma-Aldrich | Cat# P1585 | 50 ng/ml |
| Chemical compound, drug | Ionomycin | Sigma-Aldrich | Cat# I3909 | 1 μg/ml |
| Peptide | MOG35-55 | ChinaPeptides | N/A | 2 mg/ml |
| Software, algorithm | GraphPad Prism v8 | GraphPad | RRID:SCR_002798 | https://www.graphpad.com/ |
| Software, algorithm | FlowJo v10 | TreeStar | RRID:SCR_008520 | https://www.flowjo.com/flowjo/overview |
| Software, algorithm | R version v4.0.2 | R Core | RRID:SCR_001905 | http://www.r-project.org/ |
| Software, algorithm | Adobe Illustrator | Adobe | RRID:SCR_010279 | https://www.adobe.com/products/illustrator.html |

## Mice

All mice used in this study were bred for a minimum of seven generations on a C57BL/6 background. Mouse experiments, or cells from mice of the same genotype, compared littermates or age-matched control animals. The *Cxxc1*$^{fl/fl}$ mouse strain has been previously described (*Cao et al., 2016*). The *Foxp3*$^{YFP-Cre}$ mice (JAX,016959) were generously provided by Bin Li (Shanghai Jiao Tong University School of Medicine, Shanghai, China). CD45.1 (NM-KI-210226) mice were purchased from the Nanjing Biomedical Research Institute of Nanjing University. *Rag1*$^{-/-}$ mice (stock# T004753) were purchased from GemPharmatech. 2D2 (MOG35-55-specific TCR transgenic) mice were graciously supplied by Prof. Linrong Lu (Zhejiang University School of Medicine, Hangzhou, Zhejiang, China). In our study, *Foxp3*$^{YFP-Cre}$ (WT) and *Foxp3*$^{YFP-Cre}$*Cxxc1*$^{fl/fl}$ (cKO) mice, which were sex matched, were used at 3 weeks of age unless otherwise specified. The numbers of mice per experimental group are indicated in the figure legends. All mice were housed in the Zhejiang University Laboratory Animal Center under specific pathogen-free conditions, and all animal experimental procedures were approved by the Zhejiang University Animal Care and Use Committee (approval no.ZJU20230246).

## Cell culture

HEK 293T cells (ACS-4500) were obtained from ATCC, and Plat E cells were kindly provided by Prof. Xiaolong Liu (Shanghai Institutes for Biological Sciences). Their identity has been authenticated by the supplier and regular mycoplasma checks were performed. Both cell lines were cultured in Dulbecco-modified Eagle medium (DMEM) containing 10% (vol/vol) fetal bovine serum (FBS), supplemented with 1% penicillin/streptomycin.

## Immunofluorescence staining

As previously described (*Zhou et al., 2023*), coverslips were treated with a 0.01% poly-L-lysine solution (P4707; Sigma) for 10 min, air-dried, and then coated with CD4$^+$ YFP$^+$ T$_{reg}$ cells. The cells were

then fixed in 4% formaldehyde for 15 min at room temperature, permeabilized with 0.2% Triton X-100, and blocked with 1% BSA. Antibodies against CXXC1 (ab198977; Abcam) and FOXP3 (17-5773-82; Invitrogen) were diluted in Image iT FX signal enhancer (I3693; Invitrogen) and incubated with cells overnight at 4°C. After washing with phosphate-buffered saline (PBS), the cells were incubated with a goat anti-rabbit antibody Alexa Fluor 594 (1:250;10015289; Invitrogen) secondary antibody and stained with DAPI (200 ng/ml; D523; Dojindo). Slides were washed with PBS and sealed with an anti-fade solution (P36934; Invitrogen) before imaging with an Olympus FV3000 fluorescence microscope. The images were visualized using the FV31-SW software.

## Co-immunoprecipitation and western blot

Harvest the appropriate transfected cell lines and primary cells from culture and wash them with ice-cold PBS. Lyse the cells in NETN300 buffer (300 mM NaCl, 0.5 mM EDTA, 0.5% (vol/vol) NP-40, 20 mM Tris-HCl pH 8.0) supplemented with a protease inhibitor (1:100, P8340, Sigma-Aldrich) and PMSF (1 mM) on ice for 10 min. Take a small portion of the whole-cell lysate as input, and incubate the remaining lysate with either Anti-FLAG M2 Beads (M8823; Sigma) or Anti-HA Beads (HY-K0201; MCE) on a rotator at 4°C overnight. For the endogenous Co-IP assay targeting CXXC1 and FOXP3, incubate the cell lysate with protein G magnetic beads along with anti-CXXC1 (ab198977; Abcam) or anti-FOXP3 (14-4774-82; Invitrogen) antibodies on a rotator at 4°C overnight. Wash the beads three times with IP buffer (100 mM NaCl, 0.5 mM EDTA, 0.5% (vol/vol) NP-40, 20 mM Tris-HCl pH 8.0) to remove non-specific binding. Boil the washed beads with 1× Laemmli sample buffer (1610747; Bio-Rad) to elute the bound proteins. Separate the denatured proteins by SDS–PAGE. Transfer the separated proteins onto PVDF membranes (Millipore) for immunoblotting.Immunoblot the PVDF membranes (IPVH00010) with the following antibodies: anti-CXXC1 (1:1000; ab198977; Abcam), anti-FOXP3 (1:500; 14-7979-80; Invitrogen), anti-FlAG (1:1000; 14793; Cell Signaling Technology), anti-HA (1:1000; 3724S; Cell Signaling Technology). Detect the immunoblotted proteins using a secondary HRP-conjugated goat anti-rabbit antibody (1:1000; HA1001-100; Huabio) and visualize the bands using an appropriate detection method.

## ELISA

Serum samples from 3-week-old WT and KO mice were analyzed for total IgG and IgE concentrations using ELISA kits (88-50630-88; eBioscience) according to the manufacturer's instructions. Half-area ELISA plates were coated with Coating Buffer and incubated overnight at 4°C. After washing with PBST (PBS, 1 mM EDTA, 0.05% Tween-20), the plates were blocked with 5% BSA in PBS for 30 min at room temperature. Serum was diluted to the appropriate concentration with blocking buffer and incubated overnight at 4°C. After washing with PBST, the plates were incubated with HRP-conjugated anti-mouse IgG (1033-05; SouthernBiotech) and IgE (1110-05; SouthernBiotech) antibodies (1:2000 in 1% BSA/PBST) at 37°C for 1 hr. Following washing, TMB substrate was added, and the reaction was stopped with 2 M $H_2SO_4$ after sufficient color development (1–15 min). Absorbance at 450 nm was measured within 30 min.

## Lymphocyte isolation and flow cytometry

Cells from lymphoid organs were prepared by mechanical disruption between frosted slides, while non-lymphoid organs were processed enzymatically. For lung tissue, minced samples were digested in RPMI containing 100 μg/ml DNase I (9003-98-9; Sigma-Aldrich) and 2 mg/ml Collagenase D (LS004188; Worthington Biochemical) at 37°C for 1.5 hr. Liver tissue was minced and digested in RPMI supplemented with 100 μg/ml DNase I and 1 mg/ml Collagenase D at 37°C for 30 min, with lymphocytes isolated using a 40–70% Percoll (GE Healthcare) gradient. For intestinal tissue, the samples were first incubated in DMEM containing 3% FBS, 0.2% HEPES, 0.5 M EDTA, and 0.145 mg/ml dithiothreitol for 10 min. This was followed by digestion with 50 mg/ml DNase I and 145 mg/ml Collagenase II (Worthington Biochemical) in DMEM at 37°C for 5 min. Lymphocytes were then isolated using an 80% and 40% Percoll gradient.

For surface marker analysis, cells were incubated for 15 min with purified anti-mouse CD16/32 antibody (101320; BioLegend) to block Fc receptors. After blocking, cells were stained with the indicated antibodies for surface markers. To determine cytokine expression, cells were stimulated for 4 hr at 37°C with phorbol12-myristate13-acetate (50 ng/ml; S1819; Beyotime), ionomycin (1 mg/ml; S1672;

Beyotime), and brefeldin A (BFA;00-4506-51; Invitrogen). After stimulation, cells were labeled with a fixable viability dye and surface markers. Cells were then fixed and permeabilized according to the manufacturer's instructions (00-8222-49; Invitrogen). For transcription factor staining, samples were fixed using the FOXP3/Transcription Factor Staining Buffer Set (00-5523-00; Invitrogen). Flow cytometry was conducted using a BD Fortessa (BD Biosciences) or LongCyte (Beijing Challen Biotechnology Co, Ltd) system. Flow cytometry data were acquired and analyzed using FlowJo software.

The following antibodies were purchased from Invitrogen or BioLegend: Zombie Violet fixable viability (423113), Zombie NIR fixable (423105), CD25 (PC61), CD8α (53-6.7), CD62L (MEL-14), PD-1 (J43), CD44 (IM7), IL-17A (TC1118H10), KL-RG1 (2F1), CD4 (GK1.5), TCRβ (H57-597), IFN-γ (XMG1.2), FOXP3 (FJK-16s), CTLA-4 (UC10-4B9), ICOS (15F9), GITR (DTA-1), TCRVβ11 (RR3-15), CD45RB (C363-16A) CD69 (H1.2F3), T-bet (eBio4B10), IL-4 (11B11), and CD73 (TY/11.8).

## Real-time PCR

Total RNA was extracted from $T_{reg}$ cells using the RNAAiso Plus (9109; Takara) reagent according to the manufacturer's instructions, and cDNA synthesis was performed using the Prime Script RT Reagent Kit (Takara). TB Green Premix Ex Taq (RR420A; Takara) was used for quantitative real-time PCR (qPCR). The expression levels of target mRNA were normalized to the level of *Actb* expression. The primers for qPCR are as follows:

> *Cxxc1* qPCR Forward: ATCCGGGAATGGTACTGTCG
> *Cxxc1* qPCR Reverse: CTGTGGAGAAGATTTGTGGG
> *Actb* qPCR Forward: CTGTCCCTGTATGCCTCTG
> *Actb* qPCR Reverse: ATGTCACGCACGATTTCC

## CD4$^+$T and YFP$^+$ T$_{reg}$ cells adoptive transfer in EAE

CD4$^+$YFP$^+$ T$_{reg}$ cells from *Foxp3*$^{YFP-Cre}$ and *Foxp3*$^{YFP-Cre}$*Cxxc1*$^{fl/fl}$ mice were enriched using the Mouse CD4 T Cell Isolation Kit (480005; BioLegend) and then sorted using the BD Aria II flow cytometer. Naive CD4$^+$T cells from 2D2 (MOG35-55-specific TCR transgenic) mice were isolated using the Mouse CD4 Naive T cell Isolation Kit480039 (480039; BioLegend). As previously described (*Chou et al., 2021*), 2D2 naive CD4$^+$ T cells alone (5 × 10$^5$ per mouse), or 2D2 naive CD4$^+$ T cells (5 × 10$^5$ per mouse) together with WT or *Foxp3*$^{YFP-Cre}$*Cxxc1*$^{fl/fl}$ T$_{reg}$ cells (2 × 10$^5$ per mouse), were transferred into *Rag1*$^{-/-}$ mice via the tail vein. One day after cell transfer, the recipient mice were inoculated subcutaneously (s.c.) with 200 μg MOG35-55 peptide (MEVGWYRSPFSRVVHLYRNGK; GenemeSynthesis) emulsified in complete Freund's adjuvant (F5506; Sigma). Intravenous administration of 200 ng of Pertussis toxin (181; List Biological Laboratories) was performed on days 0 and 2 after peptide inoculation. The severity of EAE was monitored and blindly graded using a clinical score from 0 to 5: 0, no clinical signs; 1, limp tail; 2, paraparesis (weakness, incomplete paralysis of one or two hind limbs); 3, paraplegia (complete paralysis of two hind limbs); 4, paraplegia with forelimb weakness or paralysis; 5, dying or death.

## Isolation lymphocytes from the CNS

On day 14 after EAE induction, mice were perfused with transcardially administered PBS to eliminate contaminating blood cells in the CNS. The forebrain and cerebellum were dissected to expose the spinal cord, which was then carefully removed from the spinal canal. The fresh spinal cord was harvested and cut into 2 mm pieces. The CNS tissue pieces were homogenized using a syringe and passed through a 70-μM cell strainer to obtain a single-cell suspension. The single-cell suspension was digested with collagenase D (2 μg/ml; 11088858001; Roche) and deoxyribonuclease I (DNase I; 1 μg/ml; DN25; Sigma-Aldrich) at 37°C for 20 min under rotation. After digestion, the cell suspension was centrifuged to pellet the cells. The cell pellets were resuspended in 40% Percoll and layered onto a discontinuous Percoll gradient. Centrifugation at 80% Percoll allowed for the separation of cells at the 40–80% Percoll interface, which were collected as CNS mononuclear cells. The collected CNS mononuclear cells were washed with PBS to remove any remaining Percoll. CNS mononuclear cells were stimulated for 4 hr with PMA and ionomycin in the presence of Brefeldin A to induce cytokine production. After stimulation, cells were fixed, rendered permeable, and stained with appropriate antibodies for intracellular cytokine detection.

## Histological analyses

The lungs, skin, liver, and colon were excised from 3-week-old mice. Prior to histological analysis, the samples were fixed in formalin, embedded in paraffin, and stained with hematoxylin and eosin (H&E). For CNS histology, spinal cords were fixed in 4% paraformaldehyde, paraffin-embedded, sectioned, and stained with Luxol Fast Blue and H&E. To examine colon histology, colons from $Rag1^{-/-}$ hosts were similarly processed and stained with H&E.

## Adoptive transfer colitis model

Colitis was induced following the protocol described (**Zeng et al., 2013**). In brief, CD4$^+$ YFP$^+$ T$_{reg}$ cells were isolated from 3-week-old CD45.2$^+$ $Foxp3^{YFP-Cre}Cxxc1^{fl/fl}$ and CD45.2$^+$ $Foxp3^{YFP-Cre}$ mice. A total of $2 \times 10^5$ T$_{reg}$ cells from each group were mixed with $4 \times 10^5$ T$_{eff}$ cells (CD45.1$^+$CD4$^+$CD45RB$^{hi}$) sorted from CD45.1$^+$ mice and transferred into the $Rag1^{-/-}$ mice via intraperitoneal injection. T$_{eff}$ cells alone were transferred as a control group. Mouse body weight was measured weekly post-adoptive transfer. The percentage change in body weight was calculated by comparing the current weight with the initial weight on day 0. Mice were euthanized when any had reached 80% of their initial body weight. The large intestines were sectioned into 4 µm thick slices and stained with hematoxylin.

## In vitro T$_{reg}$ suppression assay

Naive CD4$^+$ T cells isolated from WT mice were labeled with CFSE (C34554; Invitrogen). CD4$^+$YFP$^+$ T$_{reg}$ cells from $Foxp3^{YFP-Cre}$ and $Foxp3^{YFP-Cre}Cxxc1^{fl/fl}$ mice were cultured with naive CD4$^+$ T cells (1 × 10$^5$ cells) at various ratios in the presence of 2 µg/ml anti-CD3 (16-0031-85; Invitrogen) and 3 µg/ml anti-CD28 (16-0281-85; Invitrogen). On day 3, cells were analyzed by flow cytometry.

## CUT&Tag

CUT&Tag assays of CD4$^+$YFP$^+$ T$_{reg}$ cells were conducted as previously described (**Dan et al., 2020**). Briefly, approximately 1 × 10$^5$ single cells were carefully pipetted into wash buffer twice. The pelleted cells were resuspended in wash buffer, activated concanavalin (BP531; Bangs Laboratories), and incubated for 15 min at room temperature. Cells bound to the beads were resuspended in Dig-Wash Buffer and incubated with a 1:50 dilution of primary antibodies (rabbit anti-H3K4me3, Active Motif,39016; rabbit anti-CXXC1, abcam, ab198977; rabbit anti-FOXP3, abcam, ab 150743;normal IgG, Cell Signaling, 2729) at 4°C overnight. The beads were incubated with a secondary antibody (goat anti-rabbit IgG; SAB3700883; Sigma-Aldrish) diluted 1:100 in Dig-Wash buffer for 60 min at room temperature. Cells were treated with Hyperactive pG-Tn5 Transposase (S602; Vazyme) diluted in Dig-300 Buffer for 1 hr at room temperature. The cells were subsequently resuspended in Tagmentation buffer (10 mM MgCl$_2$ in Dig-300 Buffer) and incubated at 37°C for 1 hr. To halt tagmentation, 10 µl was spiked with 0.5 M EDTA, 3 µl with 10% SDS, and 3 µl with 20 mg/ml Proteinase K and incubated at 55°C for 1 hr. DNA library amplification was performed according to the manufacturer's instructions and purified using VAHTS DNA Clean Beads (N411; Vazyme). Libraries were sequenced on the Illumina NovaSeq platform (Annoroad Gene Technology).

## CUT&Tag and ChIP-seq data analysis

FOXP3 ChIP-seq data was obtained from GSE121279. H3K27me3 ChIP-seq data was obtained from GSE14254. CUT&Tag and ChIP-seq reads were trimmed to 50 bp and aligned against the mouse genome build mm9 using Bowtie2 (v2.3.4.1) with default parameters. All PCR duplicates and unmapped reads were removed. Peak calling was performed using MACS2 (v2.1.1.20160309) and signal tracks for each sample were generated using the 'wigToBigWig' utility of UCSC. We classified the H3k4me3 peaks around TSSs into three groups: broad (>5 kb), medium (1–5 kb), and narrow (<1 kb). The top 5% of the widest peaks were considered as broad peaks. The average intensity profiles were generated using deepTools (v2.5.4). Motif analysis was performed using the 'findmotifsGenome.pl' command inHomer2 package. Epigenetic factors were identified using the Epigenetic Factor Database (https://epifactors.autosome.org/) and then screened for those that exclusively regulate the expression of their target genes by modulating the deposition of H3K4me3. Genomic distribution was analyzed using the 'genomation' R package. GO pathway analysis was performed using the 'clusterProfiler' R package. The sequencing information of CUT&Tag data generated in this study is summarized in **Supplementary file 1a**.

## Clustering analysis

Promoters were defined as ±2 kb regions flanking the annotated TSS. Reads in promoters were counted using the 'coverage' command in bedtools (v2.26.0) and further normalized to RPKM. The k-means clustering of H3K4me3 and H3K27me3 enrichment at promoters was conducted using the 'kmeans' function in R.

## RNA-seq and data analysis

Total RNA was extracted from sorted CD4+YFP+ $T_{reg}$ cells isolated from both het-WT and het-KO mice using the RNeasy Plus Mini Kit (QIAGEN, #74134), following the manufacturer's protocol. RNA-seq libraries were constructed and sequenced by Haplox (Nanchang, China), using an Illumina platform with paired-end reads of 150 bp. RNA-seq data of WT $T_{reg}$ cells was obtained from GSE82076. Raw reads were trimmed to 50 bp and mapped to the mouse genome (mm9) using TopHat (v2.1.1) with default parameters. Only uniquely mapped reads were kept for downstream analysis. The RNA abundance of each gene was quantified using Cufflinks (v2.2.1). For het-WT and het-KO RNA-seq data, gene counts were generated using HTSeq-count. For each sample, the gene count matrices were merged together and then the 'Trimmed Mean of M values' normalization (TMM) method was used to calculate the normalized expression. p values were generated using 'edgeR' R package. RNA-seq data generated in this study is summarized in *Supplementary file 1b*.

## Single-cell RNA-sequencing

A total of 300,000 sort-purified CD4+YFP+ $T_{reg}$ cells from *Foxp3*[YFP-Cre] and *Foxp3*[YFP-Cre]*Cxxc1*[fl/fl] were resuspended in BD Pharmingen Stain Buffer (FBS) (554656; BD). Single cells were isolated using a chromium controller (BD platform, BD Bioscience) according to the manufacturer's instructions, as previously described (*Chen et al., 2023*). The single cells were labeled with sample tags using the BD Mouse Immune Single-Cell Multiplexing Kit (633793; BD). Following standard protocols, cDNA amplification and library construction were performed to generate scRNA-seq libraries.

## Targeted scRNA-seq data processing

The raw FASTQ files were processed by BD Rhapsody using the Targeted analysis pipeline. After alignment and filtering, the distribution-based error correction-adjusted molecules were loaded into R Studio (version 4.3.2). All subsequent analyses were performed using the package Seurat (version 4.4.0) with default parameters. Specifically, the scRNA-seq data counts were log-normalized. All targeted genes were scaled and then were used for principal components analysis. The batch effects were removed by the HarmonyMatrix function in the Harmony package (version 1.1.0). The first 20 principal components were used to calculate nonlinear dimensionality reduction using RunUMAP. DGEs between clusters was assessed using the FindAllMarkers function. The clusters were then annotated based on DGEs. Barplots were generated using ggplot2 (version 3.4.4). Heatmaps were generated using pheatmap (version 1.0.12).

## Analysis of the single-cell TCR-seq repertoire

Raw V(D)J fastq reads were processed using BD Rhapsody Pipeline and then were analyzed using scRepertoire (version 1.12.0). The TCR clonotype was called using the nucleotide sequence of the CDR3 region for both TCR alpha and beta chains. For cells with multiple chains, the top two clonotypes with the highest expression were selected for downstream analysis. Clonal overlap between different cell types was calculated using the clonalOverlap function of scRepertoire. A clonotype was defined as expansion if it could be detected in at least two cells.

## scRNA-seq trajectory analysis

UMAP embeddings obtained from the Seurat package were projected into the Slingshot (version 2.10.0) package to construct pseudotime Trajectories for $T_{reg}$ cells. Naive subsets were set as the root state.

## WGBS and data analysis

Sorted CD4+YFP+ $T_{reg}$ cells ($3 \times 10^6$) were lysed in cell lysis buffer to release DNA. The bisulfite-treated DNA was used to prepare the sequencing library. DNA libraries were transferred to the Illumina

Platform for sequencing using 150 bp paired-end reads. Raw reads were trimmed using TrimGalore (v0.4.4) with default parameters. Subsequently, the reads were mapped against the mm9 reference genome using Bismark v0.19.0 with parameters '--bowtie2'. PCR duplicates were removed and the methylation levels were calculated using 'bismark_methylation_extractor'. We calculated the mean CpG methylation levels of various genome elements: promoter, 5'-UTR, exon, intron, 3'-UTR, gene-body, intergenic, CGIs, and repeats using in-house scripts. The sequencing information of WGBS data generated in this study is summarized in *Supplementary file 1c*.

## Statistical analysis

The statistical significance analysis was performed using Prism 8.0 (GraphPad). Error bars are presented as mean ± SD. p values of <0.05 were deemed statistically significant (*p < 0.05, **p < 0.01, ***p < 0.001, and ****p < 0.0001). Statistical analyses were performed with an unpaired *t*-test, multiple unpaired *t*-tests, or two-way ANOVA and Holm–Sidak post hoc test.

## Acknowledgements

We thank B Li (Shanghai Jiao Tong University School of Medicine, Shanghai, China) for providing *Foxp3*[YFP-Cre] mice, XY Zhou (Chinese Academy of Science (CAS), Beijing, China) for providing the FOXP3 CUT&Tag antibody, and XL Liu (Shanghai Institutes for Biological Sciences, Chinese Academy of Sciences) for his generous gifts of cell lines. We thank Y Huang, Y Li, J Wan, C Guo, and Z Lin from the Core Facilities, Zhejiang University School of Medicine for their technical support and S Hong, Y Ding, H Jin, Q Wang, and X Zhang from Animal Facilities, Zhejiang University, for feeding the mice. This work was supported by grants from the National Natural Science Foundation of China (32341002, 32030035, 32321002, 32100693, and 32270839), the National Key R&D Program of China (2023YFA1800202, 2024YFF0728703, 2022YFA1103702, and 2022YFA1103200), the Zhejiang Provincial Natural Science Foundation of China (LZ21C080001 and LZ23C070003), Science and Technology Innovation 2030-Major Project (2021ZD0200405), Key project of the Experimental Technology Program of Zhejiang University (SZD202203), and the Fundamental Research Funds for the Central Universities (226-2024-00161).

## Additional information

### Funding

| Funder | Grant reference number | Author |
| --- | --- | --- |
| National Natural Science Foundation of China | 32341002 | Lie Wang |
| National Natural Science Foundation of China | 32030035 | Lie Wang |
| National Natural Science Foundation of China | 32321002 | Li Shen |
| National Natural Science Foundation of China | 32100693 | Xiaoyu Meng |
| National Natural Science Foundation of China | 32270839 | Li Shen |
| National Key Research and Development Program of China | 2023YFA1800202 | Lie Wang |
| National Key Research and Development Program of China | 2024YFF0728703 | Lie Wang |
| National Key Research and Development Program of China | 2022YFA1103702 | Li Shen |

| Funder | Grant reference number | Author |
| --- | --- | --- |
| National Key Research and Development Program of China | 2022YFA1103200 | Li Shen |
| Natural Science Foundation of Zhejiang Province | LZ21C080001 | Lie Wang |
| Natural Science Foundation of Zhejiang Province | LZ23C070003 | Li Shen |
| National Science and Technology Major Project | 2021ZD0200405 | Lie Wang |
| Zhejiang University | SZD202203 | Lie Wang |
| Central Universities in China | 226-2024-00161 | Lie Wang |

The funders had no role in study design, data collection, and interpretation, or the decision to submit the work for publication.

## Author contributions

Xiaoyu Meng, Data curation, Formal analysis, Funding acquisition, Validation, Investigation, Visualization, Methodology, Writing - original draft, Project administration; Yezhang Zhu, Kuai Liu, Data curation, Software, Formal analysis; Yuxi Wang, Xiaoqian Liu, Chenxin Liu, Yan Zeng, Shuai Wang, Xianzhi Gao, Xin Shen, Jing Chen, Sijue Tao, Qianying Xu, Linjia Dong, Investigation, Methodology; Li Shen, Lie Wang, Conceptualization, Supervision, Funding acquisition, Project administration, Writing – review and editing

## Author ORCIDs

Li Shen ⓘ https://orcid.org/0000-0002-5696-2191
Lie Wang ⓘ https://orcid.org/0000-0001-5094-012X

## Ethics

This study was performed in strict accordance with the recommendations in the Guide for the Care and Use of Laboratory Animals of Zhejiang University. All of the animals were handled according to approved Institutional Animal Care and Use Committee (IACUC) protocols of Zhejiang University. The protocol was approved by the Committee on the Ethics of Animal Experiments of Zhejiang University (AP CODE: ZJU20230246). All surgery was performed under sodium pentobarbital anesthesia, and every effort was made to minimize suffering.

Reviewer #1 (Public review): https://doi.org/10.7554/eLife.103417.3.sa1
Reviewer #2 (Public review): https://doi.org/10.7554/eLife.103417.3.sa2
Reviewer #3 (Public review): https://doi.org/10.7554/eLife.103417.3.sa3
Author response https://doi.org/10.7554/eLife.103417.3.sa4

# Additional files

## Supplementary files

Supplementary file 1. Quality control metrics for multi-omics datasets: 1a (CUT&Tag), 1b (RNA-seq), and 1c (WGSB).

MDAR checklist

## Data availability

Sequencing data have been deposited in GEO under accession codes GSE256436 and GSE254883. All scripts used in these analyses have been made publicly available on GitHub (copy archived at *Shen, 2025*). All data generated or analyzed during this study are included in the manuscript and supporting files; source data files have been provided.

The following datasets were generated:

| Author(s) | Year | Dataset title | Dataset URL | Database and Identifier |
|---|---|---|---|---|
| Meng X, Zhu Y, Shen L, Wang L | 2025 | CxxC-finger protein 1 regulates Treg fragility associated with H3K4me3 breadth | https://www.ncbi.nlm.nih.gov/geo/query/acc.cgi?acc=GSE256436 | NCBI Gene Expression Omnibus, GSE256436 |
| Meng X, Zhu Y, Liu K, Wang L | 2025 | Single-cell transcriptomic analysis Treg cells in peripheral lymph nodes | https://www.ncbi.nlm.nih.gov/geo/query/acc.cgi?acc=GSE254883 | NCBI Gene Expression Omnibus, GSE254883 |

The following previously published datasets were used:

| Author(s) | Year | Dataset title | Dataset URL | Database and Identifier |
|---|---|---|---|---|
| Konopacki C, Pritykin Y, Rubtsov Y, Leslie C, Rudensky A | 2019 | RNA-seq and ChIP-seq to study how transcription factor Foxp1 regulates Foxp3 binding to chromatin and coordinates regulatory T cell function | https://www.ncbi.nlm.nih.gov/geo/query/acc.cgi?acc=GSE121279 | NCBI Gene Expression Omnibus, GSE121279 |
| Wei G, Wei L, Zhu J, Zang C, Hu-Li J, Yao Z, Cui K, Kanno Y, Roh T, Watford WT, Schones DE, Peng W, Sun H, Paul WE, O'Shea JJ, Zhao K | 2009 | Global Mapping of Histone H3 K4 and K27 Trimethylation: Lineage Fate Determination of Differentiating CD4+ T Cells | https://www.ncbi.nlm.nih.gov/geo/query/acc.cgi?acc=GSE14254 | NCBI Gene Expression Omnibus, GSE14254 |
| Ghosh S, Oh H | 2017 | An NF-κB-dependent, lineage-specific transcriptional program regulates Treg identity and function [RNA-seq] | https://www.ncbi.nlm.nih.gov/geo/query/acc.cgi?acc=GSE82076 | NCBI Gene Expression Omnibus, GSE82076 |

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
