## [Editor Report · eLife Assessment]

This study presents **important** findings on the role of CXXC-finger protein 1 in regulatory T cell gene regulation and function. The evidence supporting the authors' claims is **convincing**, with mostly state-of-the-art technology. The work will be of relevance to immunologists interested in regulatory T cell biology and autoimmunity.

---

## [Referee Report · Reviewer #1 (Public review)]

Summary:

This work investigated the role of CXXC-finger protein 1 (CXXC1) in regulatory T cells. CXXC1-bound genomic regions largely overlap with Foxp3-bound regions and regions with H3K4me3 histone modifications in Treg cells. CXXC1 and Foxp3 interact with each other, as shown by co-immunoprecipitation. Mice with Treg-specific CXXC1 knockout (KO) succumb to lymphoproliferative diseases between 3 to 4 weeks of age, similar to Foxp3 KO mice. Although the immune suppression function of CXXC1 KO Treg is comparable to WT Treg in an in vitro assay, these KO Tregs failed to suppress autoimmune diseases such as EAE and colitis in Treg transfer models in vivo. This is partly due to the diminished survival of the KO Tregs after transfer. CXXC1 KO Tregs do not have an altered DNA methylation pattern; instead, they display weakened H3K4me3 modifications within the broad H3K4me3 domains, which contain a set of Treg signature genes. These results suggest that CXXC1 and Foxp3 collaborate to regulate Treg homeostasis and function by promoting Treg signature gene expression through maintaining H3K4me3 modification.

Strengths:

Epigenetic regulation of Treg cells has been a constantly evolving area of research. The current study revealed CXXC1 as a previously unidentified epigenetic regulator of Tregs. The strong phenotype of the knockout mouse supports the critical role CXXC1 plays in Treg cells. Mechanistically, the link between CXXC1 and the maintenance of broad H3K4me3 domains is also a novel finding.

Weaknesses:

The authors addressed the reviewer's critiques fully in the revised manuscript.

---

## [Referee Report · Reviewer #2 (Public review)]

FOXP3 has been known to form diverse complexes with different transcription factors and enzymes responsible for epigenetic modifications, but how extracellular signals timely regulate FOXP3 complex dynamics remains to be fully understood. Histone H3K4 tri-methylation (H3K4me3) and CXXC finger protein 1 (CXXC1), which is required to regulate H3K4me3, also remain to be fully investigated in Treg cells. Here, Meng et al. performed a comprehensive analysis of H3K4me3 CUT&Tag assay on Treg cells and a comparison of the dataset with the FOXP3 ChIP-seq dataset revealed that FOXP3 could facilitate the regulation of target genes by promoting H3K4me3 deposition. Moreover, CXXC1-FOXP3 interaction is required for this regulation. They found that specific knockdown of Cxxc1 in Treg leads to spontaneous severe multi-organ inflammation in mice and that Cxxc1-deficient Treg exhibits enhanced activation and impaired suppression activity. In addition, they have also found that CXXC1 shares several binding sites with FOXP3 especially on Treg signature gene loci, which are necessary for maintaining homeostasis and identity of Treg cells.

Comments on revisions:

The authors have fully addressed the reviewers' comments and questions.

---

## [Referee Report · Reviewer #3 (Public review)]

In the report entitled "CXXC-finger protein 1 associates with FOXP3 to stabilize homeostasis and suppressive functions of regulatory T cells", the authors demonstrated that Cxxc1-deletion in Treg cells leads to the development of severe inflammatory disease with impaired suppressive function. Mechanistically, CXXC1 interacts with Foxp3 and regulates the expression of key Treg signature genes by modulating H3K4me3 deposition. Their findings are interesting and significant.

Comments on revisions:

In the revised manuscript, the authors have responded well to all the concerns reviewers raised. The manuscript has further improved.

---

## [Author Response]

The following is the authors’ response to the original reviews.

**Reviewer #1 (Public review):**
Summary:This work investigated the role of CXXC-finger protein 1 (CXXC1) in regulatory T cells. CXXC1-bound genomic regions largely overlap with Foxp3-bound regions and regions with H3K4me3 histone modifications in Treg cells. CXXC1 and Foxp3 interact with each other, as shown by co-immunoprecipitation. Mice with Treg-specific CXXC1 knockout (KO) succumb to lymphoproliferative diseases between 3 to 4 weeks of age, similar to Foxp3 KO mice. Although the immune suppression function of CXXC1 KO Treg is comparable to WT Treg in an in vitro assay, these KO Tregs failed to suppress autoimmune diseases such as EAE and colitis in Treg transfer models in vivo. This is partly due to the diminished survival of the KO Tregs after transfer. CXXC1 KO Tregs do not have an altered DNA methylation pattern; instead, they display weakened H3K4me3 modifications within the broad H3K4me3 domains, which contain a set of Treg signature genes. These results suggest that CXXC1 and Foxp3 collaborate to regulate Treg homeostasis and function by promoting Treg signature gene expression through maintaining H3K4me3 modification.Strengths:Epigenetic regulation of Treg cells has been a constantly evolving area of research. The current study revealed CXXC1 as a previously unidentified epigenetic regulator of Tregs. The strong phenotype of the knockout mouse supports the critical role CXXC1 plays in Treg cells. Mechanistically, the link between CXXC1 and the maintenance of broad H3K4me3 domains is also a novel finding.Weaknesses:(1) It is not clear why the authors chose to compare H3K4me3 and H3K27me3 enriched genomic regions. There are other histone modifications associated with transcription activation or repression. Please provide justification.

Thank you for highlighting this important point. We chose to focus on H3K4me3 and H3K27me3 enriched genomic regions because these histone modifications are well-characterized markers of transcriptional activation and repression, respectively. H3K4me3 is predominantly associated with active promoters, while H3K27me3 marks repressed chromatin states, particularly in the context of gene regulation at promoters. This duality provides a robust framework for investigating the balance between transcriptional activation and repression in Treg cells. While histone acetylation, such as H3K27ac, is linked to enhancer activity and transcriptional elongation, our focus was on promoter-level regulation, where H3K4me3 and H3K27me3 are most relevant. Although other histone modifications could provide additional insights, we chose to focus on these two to maintain clarity and feasibility in our analysis. We have revised the text accordingly; please refer to Page 18, lines 353-356.

(2) It is not clear what separates Clusters 1 and 3 in Figure 1C. It seems they share the same features.

We apologize for not clarifying these clusters clearly. Cluster 1 and 3 are both H3K4me3 only group, with H3K4me3 enrichment and gene expression levels being higher in Cluster 1. At first, we divided the promoters into four categories because we wanted to try to classify them into four categories: H3K4me3 only, H3K27me3 only, H3K4me3-H3K27me3 co-occupied, and None. However, in actual classification, we could not distinguish H3K4me3-H3K27me3 co-occupied group. Instead, we had two categories of H3K4me3 only, with cluster 1 having a higher enrichment level for H3K4me3 and gene expression levels.

(3) The claim, "These observations support the hypothesis that FOXP3 primarily functions as an activator by promoting H3K4me3 deposition in Treg cells." (line 344), seems to be a bit of an overstatement. Foxp3 certainly can promote transcription in ways other than promoting H3K3me3 deposition, and it also can repress gene transcription without affecting H3K27me3 deposition. Therefore, it is not justified to claim that promoting H3K4me3 deposition is Foxp3's primary function.

Thank you for your insightful feedback. We agree that the statement in line 344 may have overstated the role of FOXP3 in promoting H3K4me3 deposition as its primary function. As you pointed out, FOXP3 is indeed a multifaceted transcription factor that regulates gene expression through various mechanisms. It can promote transcription independent of H3K4me3 deposition, as well as repress transcription without directly influencing H3K27me3 levels.

To more accurately reflect the broader regulatory functions of FOXP3, we have revised the manuscript. The updated text (Page 19, lines 385-388) now reads:

"These findings collectively support the conclusion that FOXP3 contributes to transcriptional activation in Treg cells by promoting H3K4me3 deposition at target loci, while also regulating gene expression directly or indirectly through other epigenetic modifications.

(4) For the in vitro suppression assay in Figure S4C, and the Treg transfer EAE and colitis experiments in Figure 4, the Tregs should be isolated from Cxxc1 fl/fl x Foxp3 cre/wt female heterozygous mice instead of Cxxc1 fl/fl x Foxp3 cre/cre (or cre/Y) mice. Tregs from the homozygous KO mice are already activated by the lymphoproliferative environment and could have vastly different gene expression patterns and homeostatic features compared to resting Tregs. Therefore, it's not a fair comparison between these activated KO Tregs and resting WT Tregs.

Thank you for raising this insightful point regarding the potential activation status of Treg cells in homozygous knockout mice. To address this concern, we performed additional experiments using Treg cells isolated from *Foxp3Cre/+Cxxc1fl/fl* (hereafter referred to as “het-KO”) female mice and their littermate controls, *Foxp3Cre/+Cxxc1fl/+* (referred to as “het-WT”) mice.

The results of these new experiments are now included in the manuscript (Page25, lines 507–509, Figure 6E and Figure S6A-E):

(1) In the in vitro suppression assay, Treg cells from het-KO mice exhibited reduced suppressive function compared to het-WT Treg cells. This finding underscores the intrinsic defect in Treg cells suppressive capacity attributable to the loss of one *Cxxc1* allele.

(2) In the experimental autoimmune encephalomyelitis (EAE) model, Treg cells isolated from het-KO mice also demonstrated impaired suppressive function.

(5) The manuscript didn't provide a potential mechanism for how CXXC1 strengthens broad H3K4me3-modified genomic regions. The authors should perform Foxp3 ChIP-seq or Cut-n-Taq with WT and Cxxc1 cKO Tregs to determine whether CXXC1 deletion changes Foxp3's binding pattern in Treg cells.

Thank you for raising this important point. To address your suggestion, we performed CUT&Tag experiments and found that *Cxxc1* deletion does not alter FOXP3 binding patterns in Treg cells. Most FOXP3-bound regions in WT Treg cells were similarly enriched in KO Treg cells, indicating that *Cxxc1* deficiency does not impair FOXP3’s DNA-binding ability. These results have been added to the revised manuscript (Page 28, lines 567-575, Figure S8A-B) and are further discussed in the Discussion (Pages 28-29, lines 581-587).

**Reviewer #2 (Public review):**
FOXP3 has been known to form diverse complexes with different transcription factors and enzymes responsible for epigenetic modifications, but how extracellular signals timely regulate FOXP3 complex dynamics remains to be fully understood. Histone H3K4 tri-methylation (H3K4me3) and CXXC finger protein 1 (CXXC1), which is required to regulate H3K4me3, also remain to be fully investigated in Treg cells. Here, Meng et al. performed a comprehensive analysis of H3K4me3 CUT&Tag assay on Treg cells and a comparison of the dataset with the FOXP3 ChIP-seq dataset revealed that FOXP3 could facilitate the regulation of target genes by promoting H3K4me3 deposition.Moreover, CXXC1-FOXP3 interaction is required for this regulation. They found that specific knockdown of Cxxc1 in Treg leads to spontaneous severe multi-organ inflammation in mice and that Cxxc1-deficient Treg exhibits enhanced activation and impaired suppression activity. In addition, they have also found that CXXC1 shares several binding sites with FOXP3 especially on Treg signature gene loci, which are necessary for maintaining homeostasis and identity of Treg cells.The findings of the current study are pretty intriguing, and it would be great if the authors could fully address the following comments to support these interesting findings.Major points:(1) There is insufficient evidence in the first part of the Results to support the conclusion that "FOXP3 functions as an activator by promoting H3K4Me3 deposition in Treg cells". The authors should compare the results for H3K4Me3 in FOXP3-negative conventional T cells to demonstrate that at these promoter loci, FOXP3 promotes H3K4Me3 deposition.

Thank you for this insightful comment. We have already performed additional experiments comparing H3K4Me3 levels between FOXP3-positive Treg cells and FOXP3-negative conventional T cells (Tconv). Please refer to Pages 18, lines 361-368, and Figure 1C and Figure S1C for the results. Our results show that H3K4Me3 abundance is higher at many Treg-specific gene loci in Treg cells compared to Tconv cells. This supports our conclusion that FOXP3 promotes H3K4Me3 deposition at these loci.

(2) In Figure 3 F&G, the activation status and IFNγ production should be analyzed in Treg cells and Tconv cells separately rather than in total CD4+ T cells. Moreover, are there changes in autoantibodies and IgG and IgE levels in the serum of cKO mice?

Thank you for your valuable suggestions. In response to your comment, we reanalyzed the data in Figures 3F and 3G to assess the activation status and IFN-γ production in Tconv cells. The updated analysis revealed that *Cxxc1* deletion in Treg cells leads to increased activation and IFN-γ production in Tconv cells. Additionally, we corrected the analysis of IL-17A and IL-4 expression, which were upregulated in Tconv cells. These updated results are now included in the revised manuscript (Page 21, lines 429-431, Figure 3I and Figure S3E-F).

Additionally, we examined autoantibodies and immunoglobulin levels in the serum of *Cxxc1* cKO mice. Our data show a significant increase in serum IgG levels, accompanied by elevated IgG autoantibodies, indicating heightened autoimmune responses. In contrast, serum IgE levels remained largely unchanged. The results are detailed in the revised manuscript (Page 21, lines 421-423, Figure 3E and Figure S3B).

(3) Why did Cxxc1-deficient Treg cells not show impaired suppression than WT Treg during in vitro suppression assay, despite the reduced expression of Treg cell suppression assay -associated markers at the transcriptional level demonstrated in both scRNA-seq and bulk RNA-seq?

Thank you for your thoughtful comment. The absence of impaired suppression in *Cxxc1*-deficient Treg cells from homozygous knockout (KO) mice during the in vitro suppression assay, despite the reduced expression of Treg-associated markers at the transcriptional level (as demonstrated by scRNA-seq), can likely be explained by the activated state of these Treg cells. In homozygous KO mice, Treg cells are already activated due to the lymphoproliferative environment, resulting in gene expression patterns that differ from those of resting Treg cells. This pre-activation may obscure the effect of *Cxxc1* deletion on their suppressive function in vitro.

To address this limitation, we used heterozygous *Foxp3Cre/+Cxxc1fl/fl* (het-KO) female mice, along with their littermate controls, *Foxp3Cre/+Cxxc1fl/+* (het-WT) mice. In these heterozygous mice, we observed an impairment in Treg cell suppressive function in vitro, which was accompanied by the downregulation of several key Treg-associated genes, as confirmed by RNA-Seq analysis.

These updated findings, based on the use of het-KO mice, are now incorporated into the revised manuscript (Page 25, lines 507–509, Figure 6E).

(4) Is there a disease in which Cxxc1 is expressed at low levels or absent in Treg cells? Is the same immunodeficiency phenotype present in patients as in mice?

This is indeed a very meaningful and intriguing question, and we are equally interested in understanding whether low or absent *Cxxc1* expression in Treg cells is associated with any human diseases. However, despite an extensive review of the literature and available data, we found no reports linking *Cxxc1* deficiency in Treg cells to immunodeficiency phenotypes in patients comparable to those observed in mice.

**Reviewer #3 (Public review):**
In the report entitled "CXXC-finger protein 1 associates with FOXP3 to stabilize homeostasis and suppressive functions of regulatory T cells", the authors demonstrated that Cxxc1-deletion in Treg cells leads to the development of severe inflammatory disease with impaired suppressive function. Mechanistically, CXXC1 interacts with Foxp3 and regulates the expression of key Treg signature genes by modulating H3K4me3 deposition. Their findings are interesting and significant. However, there are several concerns regarding their analysis and conclusions.Major concerns:(1) Despite cKO mice showing an increase in Treg cells in the lymph nodes and Cxxc1-deficient Treg cells having normal suppressive function, the majority of cKO mice died within a month. What causes cKO mice to die from severe inflammation?Considering the results of Figures 4 and 5, a decrease in the Treg cell population due to their reduced proliferative capacity may be one of the causes. It would be informative to analyze the population of tissue Treg cells.

Thank you for your insightful observation regarding the mortality of cKO mice despite increased Treg cells in lymph nodes and the normal suppressive function of *Cxxc1*-deficient Treg cells.

As suggested, we hypothesized that the reduction of tissue-resident Treg cells could be a key factor. Additional experiments revealed a significant decrease in Treg cell populations in the small intestine lamina propria (LPL), liver, and lung of cKO mice. These findings highlight the critical role of tissue-resident Treg cells in preventing systemic inflammation.

This reduction aligns with Figures 4 and 5, which demonstrate impaired proliferation and survival of *Cxxc1*-deficient Treg cells. Together, these defects lead to insufficient Treg populations in peripheral tissues, escalating localized inflammation into systemic immune dysregulation and early mortality.

These additional results have been incorporated into the revised manuscript (Page21, lines 424-427, Figure 3G and Figure S3C).

(2) In Figure 5B, scRNA-seq analysis indicated that the Mki67+ Treg subset is comparable between WT and Cxxc1-deficient Treg cells. On the other hand, FACS analysis demonstrated that Cxxc1-deficient Treg shows less Ki-67 expression compared to WT in Figure 5I. The authors should explain this discrepancy.

Thank you for pointing out the apparent discrepancy between the scRNA-seq and FACS analyses regarding Ki-67 expression in *Cxxc1*-deficient Treg cells.

In Figure 5B, the scRNA-seq analysis identified the Mki67+ Treg subset as comparable between WT and *Cxxc1*-deficient Treg cells. This finding reflects the overall proportion of cells expressing Mki67 transcripts within the Treg population. In contrast, the FACS analysis in Figure 5I specifically measures Ki-67 protein levels, revealing reduced expression in *Cxxc1*-deficient Treg cells compared to WT.

To resolve this discrepancy, we performed additional analyses of the scRNA-seq data to directly compare the expression levels of Mki67 mRNA between WT and *Cxxc1*-deficient Treg cells. The results revealed a consistent reduction in Mki67 transcript levels in Cxxc1-deficient Treg cells, aligning with the reduced Ki-67 protein levels observed by FACS.

These new analyses have been included in the revised manuscript (Author response image 1) to clarify this point and demonstrate consistency between the scRNA-seq and FACS data.

**Author response image 1. sa4fig1:** Violin plots displaying the expression levels of Mki67 in T_reg_ cells from *Foxp3*^cre^ and *Foxp3*^cre^*Cxxc1*^fl/fl^ mice.

In addition, the authors concluded on line 441 that CXXC1 plays a crucial role in maintaining Treg cell stability. However, there appears to be no data on Treg stability. Which data represent the Treg stability?

Thank you for your valuable comment. We agree that our wording in line 441 may have been too conclusive. Our data focus on the impact of *Cxxc1* deficiency on Treg cell homeostasis and transcriptional regulation, rather than directly measuring Treg cell stability. Specifically, the downregulation of Treg-specific suppressive genes and upregulation of pro-inflammatory markers suggest a shift in Treg cell function, which points to disrupted homeostasis rather than stability.

We have revised the manuscript to clarify that CXXC1 plays a crucial role in maintaining Treg cell function and homeostasis, rather than stability (Page 24, lines 489-491).

(3) The authors found that Cxxc1-deficient Treg cells exhibit weaker H3K4me3 signals compared to WT in Figure 7. This result suggests that Cxxc1 regulates H3K4me3 modification via H3K4 methyltransferases in Treg cells. The authors should clarify which H3K4 methyltransferases contribute to the modulation of H3K4me3 deposition by Cxxc1 in Treg cells.

We appreciate the reviewer’s insightful comment regarding the role of H3K4 methyltransferases in regulating H3K4me3 deposition by CXXC1 in Treg cells.

CXXC1 has been reported to function as a non-catalytic component of the Set1/COMPASS complex, which includes the H3K4 methyltransferases SETD1A and SETD1B—key enzymes responsible for H3K4 trimethylation(1-4). Based on these findings, we propose that CXXC1 modulates H3K4me3 levels in Treg cells by interacting with and stabilizing the activity of the Set1/COMPASS complex.

These revisions are further discussed in the Discussion (Page 30-31, lines 624-632).

Furthermore, it would be important to investigate whether Cxxc1-deletion alters Foxp3 binding to target genes.

Thank you for raising this important point. To address your suggestion, we performed CUT&Tag experiments and found that *Cxxc1* deletion does not alter FOXP3 binding patterns in Treg cells. Most FOXP3-bound regions in WT Treg cells were similarly enriched in KO Treg cells, indicating that *Cxxc1* deficiency does not impair FOXP3’s DNA-binding ability. These results have been added to the revised manuscript (Page 28, lines 567-575, Figure S8A-B) and are further discussed in the Discussion (Pages 28-29, lines 581-587).

(4) In Figure 7, the authors concluded that CXXC1 promotes Treg cell homeostasis and function by preserving the H3K4me3 modification since Cxxc1-deficient Treg cells show lower H3K4me3 densities at the key Treg signature genes. Are these Cxxc1-deficient Treg cells derived from mosaic mice? If Cxxc1-deficient Treg cells are derived from cKO mice, the gene expression and H3K4me3 modification status are inconsistent because scRNA-seq analysis indicated that expression of these Treg signature genes was increased in Cxxc1-deficient Treg cells compared to WT (Figure 5F and G).

Thank you for your insightful comment. To clarify, the *Cxxc1*-deficient Treg cells analyzed for H3K4me3 modifications in Figure 7 were derived from *Cxxc1* conditional knockout (cKO) mice, not mosaic mice.

Regarding the apparent inconsistency between reduced H3K4me3 levels and the increased expression of Treg signature genes observed in scRNA-seq analysis (Figure 5F and G), we believe this discrepancy can be attributed to distinct mechanisms regulating gene expression. H3K4me3 is an epigenetic mark that facilitates chromatin accessibility and transcriptional regulation, reflecting upstream chromatin dynamics. However, gene expression levels are influenced by a combination of factors, including transcriptional activators, downstream compensatory mechanisms, and the inflammatory environment in cKO mice.

The upregulation of Treg signature genes in scRNA-seq data likely reflects an activated or pro-inflammatory state of *Cxxc1*-deficient Treg cells in response to systemic inflammation, as previously described in the manuscript. This contrasts with the intrinsic reduction in H3K4me3 levels at these loci, indicating a loss of epigenetic regulation by CXXC1.

To further support this interpretation, RNA-seq analysis of Treg cells from *Foxp3*^Cre/+^
*Cxxc1*^fl/fl^ (“het-KO”) and their littermate *Foxp3*^Cre/+^
*Cxxc1*^fl/+^ (“het-WT”) female mice (Figure S6C) revealed a significant reduction in key Treg signature genes such as *Icos*, *Ctla4*, *Tnfrsf18*, and *Nt5e* in het-KO Treg cells. These results align with the diminished H3K4me3 modifications observed in cKO Treg cells, further underscoring the role of CXXC1 as an epigenetic regulator.

In summary, while the gene expression changes observed in scRNA-seq may reflect adaptive responses to inflammation, the reduced H3K4me3 modifications directly highlight the critical role of CXXC1 in maintaining the epigenetic landscape essential for Treg cell homeostasis and function.

**Recommendations for the authors:**

**Reviewer #1 (Recommendations for the authors):**
In Figure 7E, the y-axis scale for H3K4me3 peaks at the Ctla4 locus should be consistent between WT and cKO samples.

We thank the reviewer for pointing out the inconsistency in the y-axis scale for the H3K4me3 peaks at the *Ctla4* locus in Figure 7E. We have carefully revised the figure to ensure that the y-axis scale is now consistent between the WT and cKO samples.

We appreciate the reviewer’s attention to this detail, as it enhances the rigor of the data presentation. Please find the updated Figure 7E in the revised manuscript.

**Reviewer #2 (Recommendations for the authors):**
In lines 455 and 466, the name of Treg signature markers validated by flow cytometry should be written as protein name and capitalized.

Thank you for pointing this out. We have carefully reviewed lines 455 and 466 and have revised the text to ensure that the Treg signature markers validated by flow cytometry are referred to using their protein names, with proper capitalization.

**Reviewer #3 (Recommendations for the authors):**
(1) On line 431, "Cxxc1-deficient cells" should be Cxxc1-deficient Treg cells".

We thank the reviewer for highlighting this oversight. On line 431, we have revised "Cxxc1-deficient cells" to "Cxxc1-deficient Treg cells" to provide a more accurate and specific description. We appreciate the reviewer's attention to detail, as this correction improves the precision of our manuscript.

(2) In Figure 4H, negative values should be removed from the y-axis.

Thank you for your observation. We have revised Figure 4H to remove the negative values from the y-axis, as requested. This adjustment ensures a more accurate and meaningful representation of the data.

(3) It is better to provide the lists of overlapping genes in Figure 7C.

Thank you for your suggestion. We agree that providing the lists of overlapping genes in Figure 7C would enhance the clarity and reproducibility of the results. We have now included the gene lists as supplementary information (Supplementary Table 3) accompanying Figure 7C.

(1) Lee, J. H. & Skalnik, D. G. CpG-binding protein (CXXC finger protein 1) is a component of the mammalian set1 histone H3-Lys4 methyltransferase complex, the analogue of the yeast Set1/COMPASS complex. Journal of Biological Chemistry 280, 41725-41731, doi:10.1074/jbc.M508312200 (2005).

(2) Thomson, J. P., Skene, P. J., Selfridge, J., Clouaire, T., Guy, J., Webb, S., Kerr, A. R. W., Deaton, A., Andrews, R., James, K. D., Turner, D. J., Illingworth, R. & Bird, A. CpG islands influence chromatin structure via the CpG-binding protein Cfp1. Nature 464, 1082-U1162, doi:10.1038/nature08924 (2010).

(3) Shilatifard, A. in Annual Review of Biochemistry, Vol 81 Vol. 81 Annual Review of Biochemistry (ed R. D. Kornberg) 65-95 (2012).

(4) Brown, D. A., Di Cerbo, V., Feldmann, A., Ahn, J., Ito, S., Blackledge, N. P., Nakayama, M., McClellan, M., Dimitrova, E., Turberfield, A. H., Long, H. K., King, H. W., Kriaucionis, S., Schermelleh, L., Kutateladze, T. G., Koseki, H. & Klose, R. J. The SET1 Complex Selects Actively Transcribed Target Genes via Multivalent Interaction with CpG Island Chromatin. Cell Reports 20, 2313-2327, doi:10.1016/j.celrep.2017.08.030 (2017).